# Nuclear receptor LRH-1/NR5A2 is required and targetable for liver endoplasmic reticulum stress resolution

Jennifer L Mamrosh[1], Jae Man Lee[1], Martin Wagner[1], Peter J Stambrook[2], Richard J Whitby[3], Richard N Sifers[1,4], San-Pin Wu[1], Ming-Jer Tsai[1], Francesco J DeMayo[1], David D Moore[1]*

[1]Department of Molecular and Cellular Biology, Baylor College of Medicine, Houston, United States; [2]Department of Molecular Genetics, Biochemistry, and Molecular Biology, University of Cincinnati, Cincinnati, United States; [3]Department of Chemistry, University of Southampton, Southampton, United Kingdom; [4]Department of Pathology and Immunology, Baylor College of Medicine, Houston, United States

**Abstract** Chronic endoplasmic reticulum (ER) stress results in toxicity that contributes to multiple human disorders. We report a stress resolution pathway initiated by the nuclear receptor LRH-1 that is independent of known unfolded protein response (UPR) pathways. Like mice lacking primary UPR components, hepatic *Lrh-1*-null mice cannot resolve ER stress, despite a functional UPR. In response to ER stress, LRH-1 induces expression of the kinase *Plk3*, which phosphorylates and activates the transcription factor ATF2. *Plk3*-null mice also cannot resolve ER stress, and restoring *Plk3* expression in *Lrh-1*-null cells rescues ER stress resolution. Reduced or heightened ATF2 activity also sensitizes or desensitizes cells to ER stress, respectively. LRH-1 agonist treatment increases ER stress resistance and decreases cell death. We conclude that LRH-1 initiates a novel pathway of ER stress resolution that is independent of the UPR, yet equivalently required. Targeting LRH-1 may be beneficial in human disorders associated with chronic ER stress.

*For correspondence: moore@bcm.edu

**Competing interests:** The authors declare that no competing interests exist.

## Introduction

It has been estimated that close to one-third of newly synthesized proteins cannot properly fold and are therefore subjected to rapid intracellular degradation (*Schubert et al., 2000*). Both the large scale of this process and the toxicity of misfolded proteins dictate that protein folding must be tightly controlled. When cellular protein folding demands in the cell exceed capacity, misfolded proteins accumulate within the endoplasmic reticulum (ER), causing ER stress. In many human diseases, especially metabolic and neurodegenerative disorders, cellular folding capacities are overwhelmed by increased protein synthesis rates and/or accumulation of abnormal proteins, resulting in ER stress-mediated cell toxicity or death (*Lin et al., 2008*).

A set of highly conserved signaling pathways collectively termed the unfolded protein response (UPR) functions to resolve ER stress. The UPR consists of three distinct pathways, each of which is initiated by a specific ER-bound protein that acts as a sensor for misfolded proteins. Activation of these sensors elicits downstream signaling pathways that ultimately produce three transcription factors, ATF4, ATF6, and XBP-1, which are active at genes involved in protein folding, expansion of the ER, protein degradation, and, in the case of severe stress, apoptosis. In recent years, many additional components of ER stress resolution have been identified, often with roles in maintenance of metabolic homeostasis following exposure to stress.

**eLife digest** A protein can only work properly if it has been folded into the correct shape. However, it is estimated that about one third of new proteins have the wrong shape. This is a major challenge for cells because misfolded proteins are often toxic, and cause many neurodegenerative and metabolic disorders.

In eukaryotic cells, most protein folding takes place inside a part of the cell called the endoplasmic reticulum (ER). If an incorrectly folded protein is detected, it is prevented from leaving the ER until it is refolded correctly, or destroyed. If too many proteins are misfolded, a process called the unfolded protein response helps the cell to cope with this 'ER stress' by expanding the ER and producing more of the molecules that assist protein folding. If this does not relieve the ER stress, the cell self-destructs. Neighboring cells then have to increase protein production to compensate for what would have been produced by the dead cell, thereby increasing the chance that they will also experience ER stress.

Activation of a protein called LRH-1 (short for liver receptor homolog-1) that is produced in the liver, pancreas and intestine can relieve the symptoms of the various metabolic diseases that are associated with chronic ER stress, including type II diabetes and fatty liver disease. However, researchers have been puzzled by the fact that although LRH-1 performs many different roles, its molecular structure provides few clues as to how it can do this.

Mamrosh et al. now confirm the speculated link between LRH-1 and ER stress relief in mice. LRH-1 triggers a previously unknown pathway that can relieve ER stress and is completely independent of the unfolded protein response. Targeting LRH-1 with certain chemical compounds alters its activity, suggesting that drug treatments could be developed to relieve ER stress. As similar targets for drugs have not been found in the unfolded protein response, the discovery of the LRH-1 pathway could lead to new approaches to the treatment of the diseases that result from ER stress.

Despite the increasing number of factors involved in the ER stress response, only a few are directly responsible for ER stress resolution. Treatment of knockout mice for any of the upstream UPR pathway components, *Atf6*, *Ire1α* (product of *Ern1*), or *Perk* (product of *Eifak3*), with tunicamycin, a chemical stressor that primarily affects the liver, elicits a striking phenotype. All knockouts exhibit an inability to resolve ER stress upon exposure, as well as a striking metabolic phenotype of strong and persistent accumulation of hepatic triglycerides (*Rutkowski et al., 2008*; *Teske et al., 2011*; *Zhang et al., 2011*). While tunicamycin is a supraphysiological stressor, these results are important because no other mouse models have been reported to share these phenotypes following ER stress, suggesting that the canonical UPR pathways are functionally overlapping, yet non-redundant, and also that the UPR contains the only required pathways for ER stress resolution.

The orphan nuclear hormone receptor liver receptor homolog-1 (LRH-1; product of *Nr5a2*) is expressed in secretory tissues or tissues with high rates of protein production, including liver, pancreas, intestine, and reproductive tissue (*Higashiyama et al., 2007*). None of the divergent roles of LRH-1— notably bile acid production, development and maintenance of pluripotency, and local steroidogenesis— directly imply a role in ER stress resolution. However, the fact that metabolic diseases such as type II diabetes and fatty liver disease are associated with chronic ER stress (*Ozcan et al., 2006*), but can be alleviated by activation of LRH-1 (*Lee et al., 2011*), suggested a potential connection. In the present study, this connection was substantiated by our discovery that LRH-1 is indispensible for ER stress resolution. In this unexpected role, LRH-1 initiates the kinase-driven activation of a CREB-like transcription factor following exposure to ER stress, specifically through polo-like kinase 3 (*Plk3*) induction of activating transcription factor 2 (ATF2) phosphorylation. This response is absent in *Lrh-1* liver-specific knockout mice, but restoration of *Plk3* induction to *Lrh-1* null cells reestablishes ER stress resolution. Similarly, expression of a constitutively active *Atf2* restores ER stress resolution capacity to *Lrh-1* null cells, and reduction of ATF2 activity sensitizes control cells to ER stress. Treatment of hepatocytes with LRH-1 agonists increases the capacity for ER stress resolution and diminishes toxicity resulting from severe ER stress. Thus, we report the existence of an unexpected kinase-driven and drug-targetable pathway responding to ER stress that lies outside of the classical UPR pathways, but is equally required for ER stress resolution.

# Results

## Loss of *Lrh-1* results in liver fat accumulation, cell death, and inability to resolve ER stress upon exposure

To test whether LRH-1 is involved in ER stress resolution, we treated *Lrh-1* liver-specific knockout (*Lrh-1$^{LKO}$*) mice and control littermates (*Lrh-1$^{f/f}$*) with the ER stress inducer tunicamycin (TM), which primarily affects the liver. *Lrh-1$^{LKO}$* mice exhibited profound hepatic lipid accumulation by 48 hr following stress, as evidenced by macroscopic evaluation (***Figure 1A***) and measurement of increased hepatic triglycerides and free fatty acids (***Figure 1B***). We also observed increased TUNEL staining by 72 hr following stress in *Lrh-1$^{LKO}$* mice, confirming that the prolonged ER stress had resulted in increased apoptosis (***Figure 1C,D***). As expected, *Lrh-1$^{LKO}$* mice exhibit increased caspase and PARP cleavage following TM (***Figure 1E***). To confirm that this response was associated with misfolded proteins, we stained primary hepatocytes from TM-treated control and *Lrh-1$^{LKO}$* mice with Thioflavin T, which fluoresces when bound to protein aggregates and therefore can be used to quantitate ER stress (***Beriault and Werstuck, 2013***). We observed little increase in staining for control cells treated with TM, suggesting that resolvable ER stress induced by TM does not result in significant protein aggregation, but observed strong staining in *Lrh-1$^{LKO}$* cells increasing over time treated with TM (***Figure 1F***).

The hepatic lipid accumulation phenotype observed in *Lrh-1$^{LKO}$* mice is identical to that observed in mice lacking any of the three UPR branches following TM treatment, suggesting that there could be a significant deficit in one or more of these pathways in our *Lrh-1$^{LKO}$* mice. Thus, we assessed the nuclear accumulation of UPR transcription factors that represent the most downstream effector of each branch. In contrast to our expectation of a UPR deficiency, we observed comparable nuclear accumulation of downstream transcription factors for all three UPR branches in response to ER stress in both control and *Lrh-1$^{LKO}$* mice, suggesting that all three branches were functional (***Figure 1G***). Target genes dependent on each of the three UPR pathways were similarly induced following TM stress in both control and *Lrh-1$^{LKO}$* mice, confirming that XBP-1, ATF6, and ATF4 were transcriptionally active in addition to being nuclear localized in *Lrh-1$^{LKO}$* mice (***Figure 1—figure supplement 1***). Importantly, however, we observed sustained signaling of these in *Lrh-1$^{LKO}$* mice, indicating that ER stress could not be resolved in *Lrh-1$^{LKO}$* mice despite a functional UPR (***Figure 1E***).

In addition to tunicamycin, we utilized two additional chemical ER stress inducers, dithiothreitol (DTT), and Brefeldin A (BFA), which induce ER stress via different mechanisms and kinetics in comparison to TM. Primary hepatocytes from control mice treated with DTT or BFA exhibit a mild or nonexistent UPR response at earlier times, and no UPR signaling at later times (***Figure 2A***). In contrast, hepatocytes from *Lrh-1$^{LKO}$* mice exhibit a more robust initial UPR response as well as sustained signaling at later times, indicating failure to resolve these stresses. Cell death in response to DTT and BFA was significantly higher in *Lrh-1$^{LKO}$* cells as compared to controls (***Figure 2B***). In addition, there is a trend towards increased fat accumulation in *Lrh-1$^{LKO}$* cells treated with BFA (***Figure 2C***), consistent with that observed in *Lrh-1$^{LKO}$* mice treated with TM (***Figure 1A,B***).

To confirm that the effect of loss of *Lrh-1* in failure to resolve ER stress is not due to indirect effects on drug metabolism of the ER stressors utilized, we also evaluated the effect of a non-chemical inducer of ER stress. Liver regeneration in response to partial hepatectomy induces transient ER stress and the response of UPR pathway-deficient mice to partial hepatectomy is similar to their response to TM treatment, in that liver fat accumulates due to unresolved ER stress (***Zhang et al., 2011***). We performed partial hepatectomy or sham surgery on control and *Lrh-1$^{LKO}$* mice and observed dramatic fat accumulation following partial hepatectomy in *Lrh-1$^{LKO}$* mice (***Figure 2D***). In addition, we assessed nuclear UPR transcription factor accumulation as a marker for ER stress. Unlike control mice, *Lrh-1$^{LKO}$* mice exhibited accumulation of these factors following partial hepatectomy (***Figure 2E***), indicating that they were not able to resolve ER stress caused by liver regeneration.

## Lrh-1 target genes are induced following ER stress, with heightened *Lrh-1* expression dependent on UPR components

Following ER stress, *Lrh-1* mRNA expression is moderately induced (***Figure 3A***), although this does not appear to be reflected in LRH-1 protein levels (***Figure 3B***). Interestingly, we still observe a TM-dependent induction of LRH-1 target genes *Cyp7a1* and *Cyp8b1* (***Figure 3C***), which are involved in bile acid biosynthesis, although no change in total hepatic bile acid in response to stress or genotype (***Figure 3D***). To determine whether components of the UPR could affect the transcriptional activity of LRH-1,

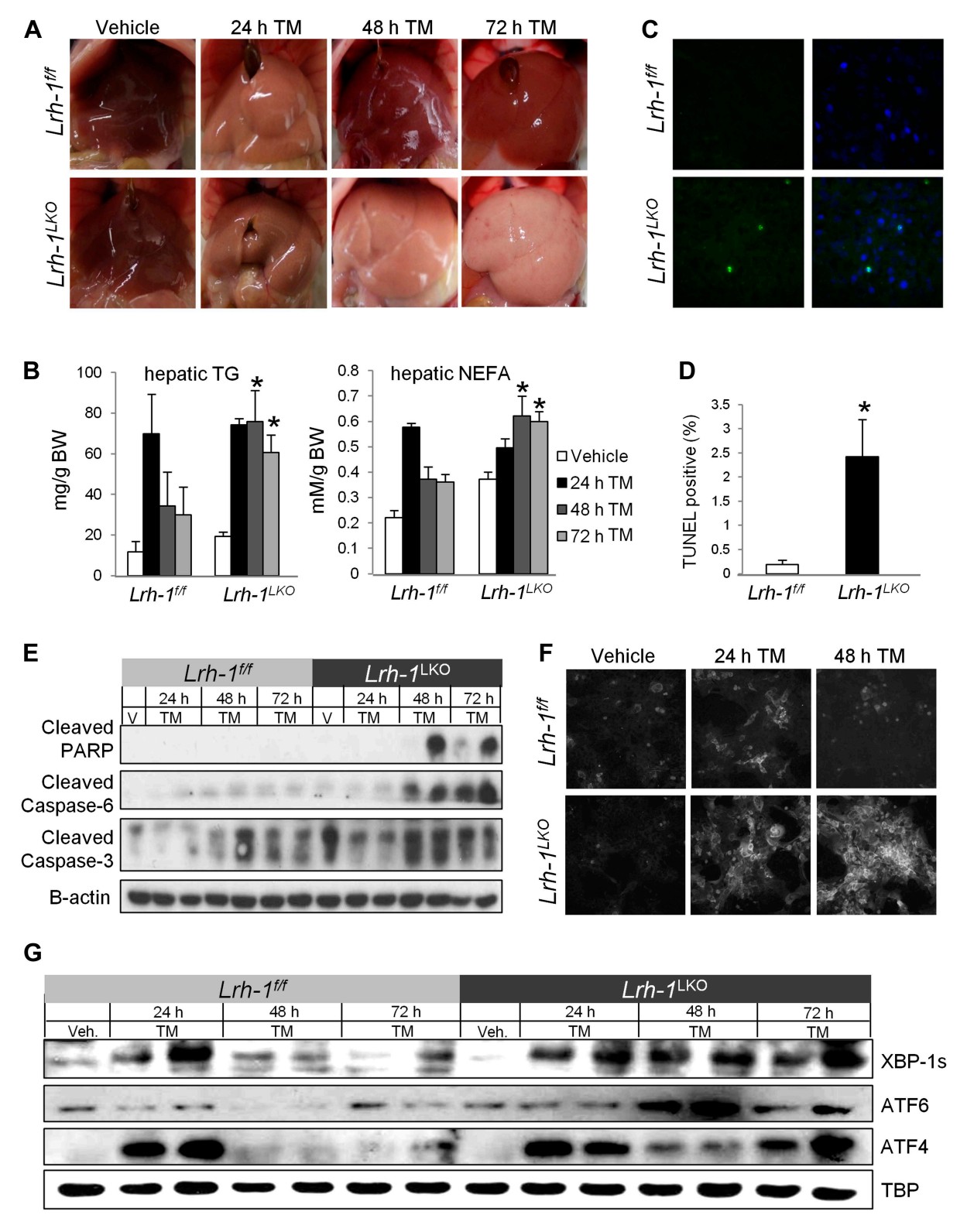

**Figure 1.** *Lrh-1* is required for ER stress resolution and for protection against stress-induced lipid accumulation and cell death. (**A**) Macroscopic visualization of steatosis following ER stress in *Lrh-1 LKO* mice. Mice were i.p. injected with 1 mg/kg tunicamycin (TM) or vehicle and livers photographed following sacrifice. Representative of 3–6 mice per group. (**B**) Quantification of hepatic triglycerides and non-esterified free fatty acids of control *(Lrh-1f/f)* and *Figure 1. Continued on next page*

*Figure 1. Continued*

*Lrh-1^LKO* mice (n = 3–6) injected with TM or vehicle. (**C**) Representative hepatic TUNEL staining for apoptosis of control and *Lrh-1^LKO* mice injected with 1 mg/kg TM and sacrificed at 72 hr. Green fluorescence represents TUNEL-positive and blue represents DAPI-positive (merged image on right). Magnification at 40x (objective). Representative of three mice per group. (**D**) Quantification of TUNEL-positive cells for control and *Lrh-1^LKO* mice treated with TM and sacrificed at 72 hr. Ratio of TUNEL-positive to DAPI-positive cells was calculated from three fields of each slide (n = 3). Significance at p<0.05. (**E**) Immunoblot of cleaved PARP, cleaved caspase 3, and cleaved caspase 6 in cytoplasmic fractions generated from control and *Lrh-1^LKO* mice (n = 3–6; pooled) injected with 1 mg/kg tunicamycin (TM) or vehicle. β-actin was used as a loading control. (**F**) Thioflavin T fluorescence of protein aggregates in primary hepatocytes prepared from control and *Lrh-1^LKO* mice, treated with vehicle or 0.01 µg/ml TM, and fixed in 4% PFA before staining with 500 µM Thioflavin T. Magnification at 10x (objective). Representative of three mice per group. (**G**) Immunoblot of nuclear spliced XBP-1, cleaved ATF6, and ATF4 for control and *Lrh-1^LKO* mice (n = 3–6; pooled) injected with 1 mg/kg tunicamycin (TM) or vehicle. TBP was used as a loading control.

The following figure supplements are available for figure 1:

**Figure supplement 1**. Loss of *Lrh-1* does not result in loss of UPR target genes in response to stress.

we used siRNA to knockdown *Ire1a*, *Perk*, and *Atf6* in primary hepatocytes from control mice. We observed robust silencing of these genes and subsequently treated cells with tunicamycin. The LRH-1 target genes *Shp* (product of *Nr0b2*) and *Plk3* (introduced below) are induced in response to ER stress, but this induction is significantly blunted following knockdown of *Ire1a*, similar to the loss of *Lrh-1* (*Figure 3E*). Knockdown of either *Ire1a* or *Atf6* also decreased the modest stress-dependent induction of *Lrh-1* mRNA expression. However, the loss of *Atf6* did not decrease induction of *Shp* by TM, indicating a specific effect of IRE1a on *Lrh-1* transactivation. It is quite unlikely that ER-bound IRE1a directly regulates LRH-1, as GFP-tagged LRH-1 is exclusively nuclear in the presence or absence of TM (*Figure 3F*).

## Stress-inducible ATF2 phosphorylation is impaired following loss of *Lrh-1*

The striking phenotype of the *Lrh-1^LKO* mice in the absence of any apparent defect in the three canonical UPR pathways predicted that LRH-1 could initiate a novel pathway of ER stress resolution. We performed microarray analysis on control and *Lrh-1^LKO* mice treated with vehicle or TM to identify LRH-1-dependent TM responsive genes, but initial analysis of the results did not identify obvious candidates that could account for the *Lrh-1^LKO* phenotype. To identify potentially relevant signaling pathways, we used the Molecular Signatures Database (MSigDB) program to identify transcription factor binding motifs that were significantly overrepresented in promoters of the top 100 genes differentially responsive to TM between genotypes. The predominant output of this analysis was the cyclic AMP response element (CRE) (*Figure 4A*). Among the multiple transcription factors that can bind this motif, activating transcription factor 2 (ATF2) was a promising candidate for further investigation based on its known roles in responding to genotoxic and oxidative stresses (*van Dam et al., 1995*; *Kurata, 2000*). *Atf2* levels were slightly lower in *Lrh-1^LKO* mice compared to controls during basal conditions, but there were no significant differences in levels of *Atf2* between genotypes following treatment with TM (*Figure 4B*). Because ATF2 requires phosphorylation to be transcriptionally active (*Livingstone et al., 1995*), we assessed its phosphorylation status. In TM-treated primary hepatocytes from control and *Lrh-1^LKO* mice, we observed a rapid induction of ATF2 phosphorylation at the residues (T51/53, corresponding to human T69/71) required for ATF2 transcriptional activity in control mice (*Figure 4C*), and this response was absent in *Lrh-1^LKO* mice.

To further link ATF2 and the *Lrh-1*-dependent ER stress response, we screened for genes that were induced at least 1.5-fold by TM in control mice with significantly different induction in *Lrh-1^LKO* mice, and secondarily screened for those with significantly different expression levels following TM treatment between genotypes. From this filtering we obtained a list of 65 genes (*Table 1*) and further investigated their capacity to be bound and regulated by ATF2. Genome-wide ATF2 binding data from the ENCODE Project (*ENCODE Project Consortium, 2011*) was used to identify genes with ATF2 binding sites −500 bp to +275 bp from the transcriptional start site (TSS). The same analysis was performed with genome-wide LRH-1 binding data (*Chong et al., 2012*). Our gene list was compared with known ATF2 and LRH-1 binding sites, and genes bound by ATF2 or LRH-1 −500 bp to +275 bp from the TSS were determined (*Table 1*). Hypergeometric distribution tests determined our list

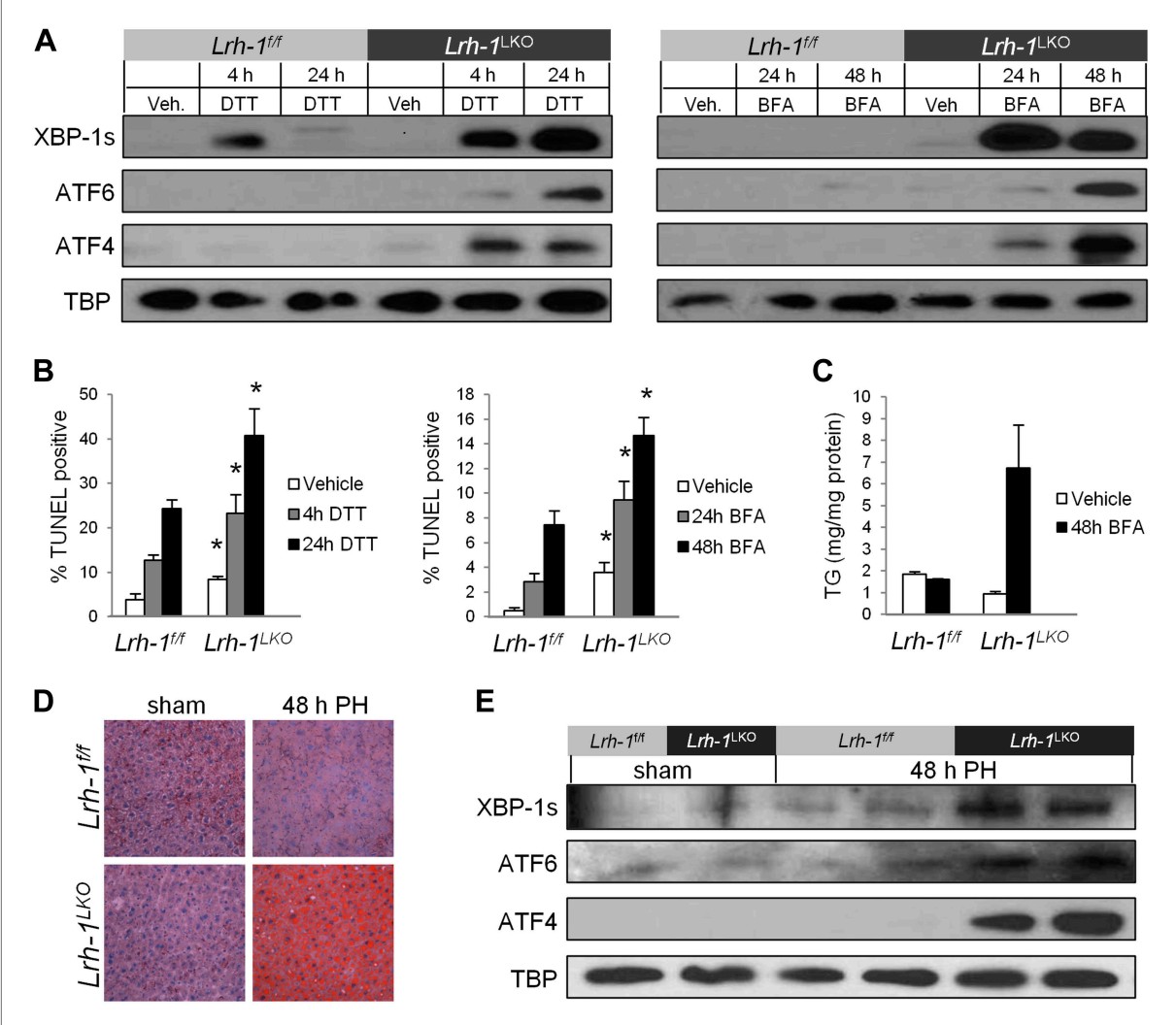

**Figure 2**. Loss of *Lrh-1* sensitizes mice to ER stress resulting from chemical and physiological inducers. (**A**) Immunoblot of nuclear spliced XBP-1, cleaved ATF6, and ATF4 for primary hepatocytes from control and *Lrh-1LKO* mice treated with 2 mM DTT or 0.05 μg/ml Brefeldin A (BFA). TBP was used as a loading control. Results representative of three independent experiments. (**B**) Quantification of TUNEL-positive cells for primary hepatocytes from control and *Lrh-1LKO* mice treated with 2 mM DTT or 0.05 μg/ml BFA. Ratio of TUNEL-positive to DAPI-positive cells was calculated from four fields of each slide (n = 3). Significance at p<0.01. (**C**) Quantification of triglycerides from primary hepatocytes from control and *Lrh-1LKO* mice (n = 3) treated with vehicle or 0.05 μg/ml BFA for 48 hr. TG was normalized to total protein by Bradford assay. (**D**) Partial hepatectomy (PH) was used as a non-chemical ER stress inducer. Control and *Lrh-1LKO* mice underwent surgical removal of 70% of liver weight or sham surgery and were sacrificed 48 hr post surgery. Oil Red O staining (200x) for neutral lipid accumulation on PH or sham surgery liver samples. Results representative of four independent samples. (**E**) Nuclear protein was extracted and immunoblotted for XBP-1s, ATF6, and ATF4 with TBP as a loading control for control and *Lrh-1LKO* mice 48 hr after sham or PH surgery. Results representative of four independent samples.

was highly enriched for ATF2-bound genes (p=3.83e−08) and enriched, although less strikingly so, for LRH-1-bound genes (p=1.40e−04).

We chose three direct ATF2 target genes—*B3gat3*, likely involved in proteoglycan synthesis (*Baasanjav et al., 2011*), *Creld1*, an ER stress-inducible protein of unknown function (*Hansen et al., 2012*), and *Mcfd2*, a receptor for ER–Golgi transport (*Zhang et al., 2003*)—for further studies. These genes were induced by TM in control mice but showed blunted induction in *Lrh-1LKO* mice (*Figure 4D*). To determine whether ATF2 is bound, phosphorylated, and enzymatically active on these genes, we performed chromatin immunoprecipitation (ChIP) for ATF2, pATF2 (m51/53; h69/71), and acetylated histone H4 (AcH4), a marker for activated ATF2 histone acetyltransferase activity (*Bruhat et al., 2007*). ChIP was performed on chromatin from control and *Lrh-1LKO* mice treated with vehicle or TM. While

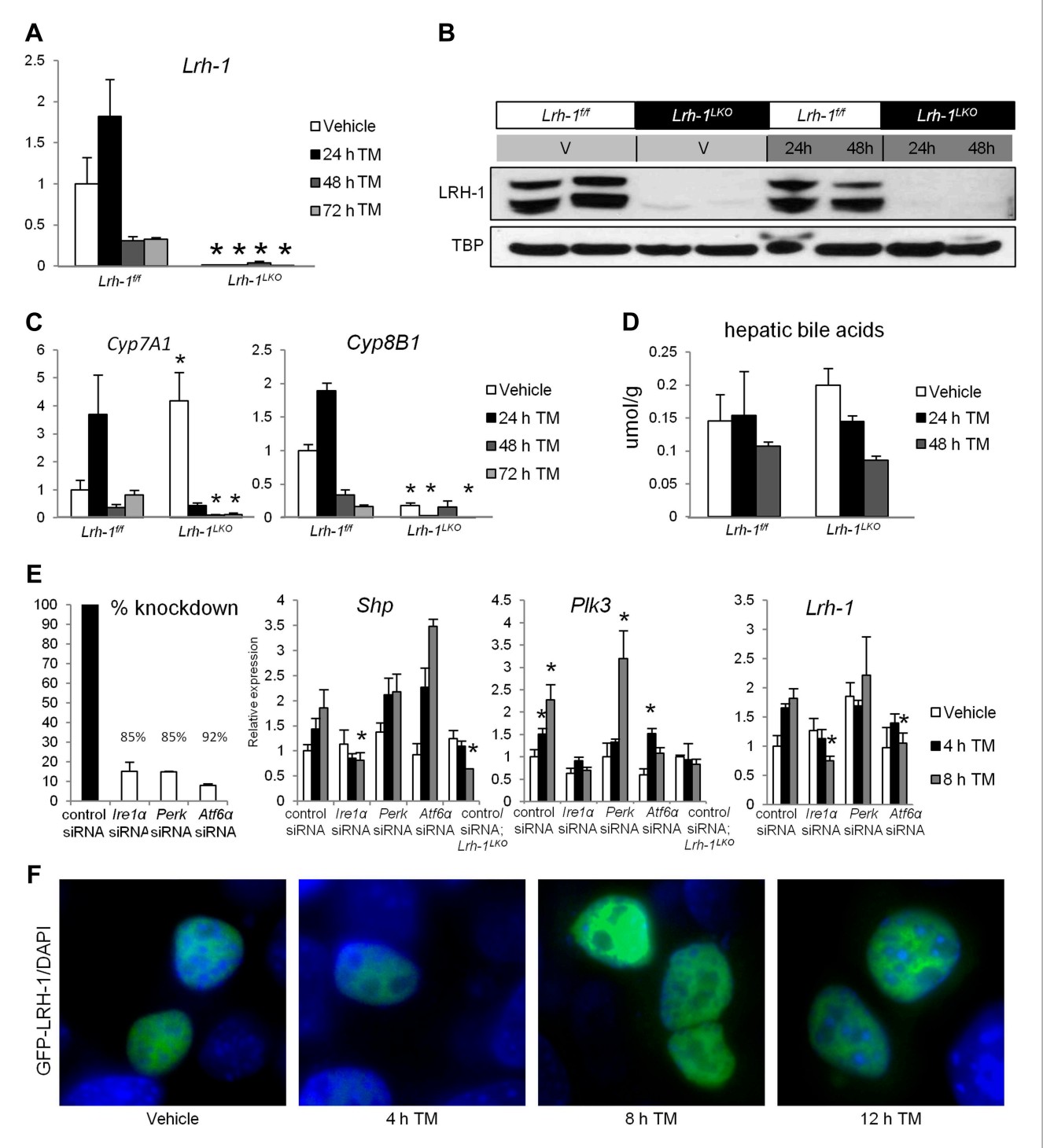

**Figure 3.** An increase in LRH-1 transcriptional activity and expression is observed following ER stress, with heightened expression dependent on UPR components. (**A**) Relative expression by quantitative PCR for *Lrh-1*. RNA was collected at designated timepoints for control and *Lrh-1^LKO^* mice treated with vehicle or 1 mg/kg tunicamycin (TM) (n = 3–6). Data were normalized to *Tbp* expression. (**B**) Western blot analysis of nuclear LRH-1 for control and *Lrh-1^LKO^* mice injected with 1 mg/kg TM or vehicle and sacrificed at designated timepoints. TBP was used as a loading control. Results representative of 3–6 individual samples. (**C**) Relative expression by quantitative PCR for *Cyp7A1* and *Cyp8B1*, which are LRH-1 target genes. RNA was collected at designated timepoints for control and *Lrh-1^LKO^* mice treated with vehicle or 1 mg/kg TM (n = 3–6). Data were normalized to *Tbp* expression. Significance at p<0.05 between genotypes. (**D**) Hepatic bile acid was extracted from tissue from control and *Lrh-1^LKO^* mice treated with vehicle or 1 mg/kg TM (n = 3–6) and sacrificed at designated timepoints. (**E**) Primary hepatocytes from control and *Lrh-1^LKO^* mice were transfected with siRNA

*Figure 3. Continued on next page*

*Figure 3. Continued*

against *Ire1a*, *Perk,* or *Atf6*, or a nonsilencing siRNA. Percent knockdown was quantified by qPCR for *Ire1a*, *Perk,* and *Atf6* in control cell samples (n = 3–4) 52 hr after transfection. Relative expression for *Shp*, *Plk3,* and *Lrh-1* was by qPCR for primary hepatocytes from control and *Lrh-1*[LKO] mice treated with vehicle or 5 ng/ml TM (n = 3–4). TBP was used as a loading control. Significance at p<0.05 as compared with control siRNA samples from control mice. (F) N-terminal-tagged GFP-LRH-1 fluorescence (green) in TLR-3 cells transfected with GFP-LRH-1 and treated with vehicle or 1 µg/ml TM for timepoints indicated. DNA was stained with DAPI (blue). Magnification at 100x (objective) and images cropped. Results representative of three independent experiments.

there was little difference in ATF2 binding between treatments or genotypes for *B3gat3*, *Creld1*, and *Mcfd2*, binding of pATF2 and AcH4 was increased following TM treatment in control mice alone for these three genes (*Figure 4E*). Overall, we conclude that a significant number of genes differentially regulated by ER stress in control and *Lrh-1*[LKO] mice are ATF2 targets dependent on activation of ATF2 by phosphorylation.

## Lrh-1 transcriptionally controls *Plk3* expression to promote ATF2 phosphorylation

We did not observe significantly reduced activation of kinases known to phosphorylate ATF2, particularly the MAP kinases JNK and p38 (*Ouwens et al., 2002*), in *Lrh-1*[LKO] mice in response to ER stress. However, polo-like kinase 3 (Plk3), which we found to be differentially induced by TM between genotypes (*Table 1*), has also been reported to be capable of phosphorylating ATF2 (*Wang et al., 2011b*), although not in the context of ER stress. In line with our microarray findings, *Plk3* mRNA expression was robustly induced in control mice following ER stress, and this induction was significantly diminished in *Lrh-1*[LKO] mice (*Figure 5A*). This ER stress-induced increase in control mice alone was also observed at the level of PLK3 protein (*Figure 5B*). A time course of *Plk3* induction performed in primary hepatocytes from control and *Lrh-1*[LKO] mice (*Figure 5C*) indicates that *Plk3* induction occurs in a similar time frame to ATF2 phosphorylation, suggesting that this induction could be sufficient for the stress-inducible ATF2 phosphorylation observed (*Figure 4C*). We predicted that *Plk3* was a direct transcriptional target of LRH-1 and identified a strong LRH-1 binding site 285 base pairs upstream of the Plk3 TSS. Chromatin immunoprecipitation of LRH-1 identified an enrichment of LRH-1 occupancy at this site shortly after TM treatment in control mice (*Figure 5D*).

## *Plk3* is required for induction of CRE-dependent genes and ER stress resolution

To determine the requirement for PLK3 in induction of CRE-containing ATF2 target genes, we utilized the cell line TLR-3, which was reported to have stress-inducible ATF2 phosphorylation in response to environmental toxins (*Muguruma et al., 2008*). We observed that ATF2 was phosphorylated in response to TM dependent on *Lrh-1* expression (*Figure 5E*). Luciferase reporter assays utilizing ATF2 and a cAMP response element reporter demonstrated an increase in reporter activity under ER stress conditions, suggesting increased ATF2 phosphorylation (*Figure 5F*). Inhibition of JNK, p38, and SGK did not significantly reduce reporter activity whereas PLK3 inhibition did, arguing that PLK3 is the predominant driver of ATF2 phosphorylation following ER stress. For endogenous gene expression, we quantified ATF2 target gene *Atf3* (*Mayer et al., 2008*) induction following TM treatment, and observed that it was significantly blunted with *Lrh-1* knockdown. However, overexpression of a constitutively active *Atf2* (C2/Atf2; *Steinmuller and Thiel, 2003*) or *Plk3* restored stress-inducible *Atf3* induction (*Figure 5G*). We then assessed *Atf3* induction in primary hepatocytes from control and *Lrh-1*[LKO] mice treated with TM or GW843682X, a PLK3 inhibitor. We observed that hepatocytes from *Lrh-1*[LKO] mice had blunted *Atf3* induction in response to stress, and this was not affected by treatment with GW843682X (*Figure 5H*). However, in control hepatocytes, the robust stress-dependent induction of *Atf3* was blunted to levels comparable to *Lrh-1*[LKO] mice when PLK3 was inhibited.

These results predict that PLK3 is critical for ER stress resolution. To address this, we first treated control mice with tunicamycin and GW843682X. PLK3 inhibition resulted in decreased ability to resolve ER stress, as evidenced by sustained UPR signaling at 48 hr (*Figure 5I*), reminiscent of unresolved ER stress observed in *Lrh-1*[LKO] mice (*Figure 1G*). We then obtained control and *Plk3*[−/−] mice (*Myer et al., 2011*) and treated them with TM. UPR transcription factors were comparably induced observed

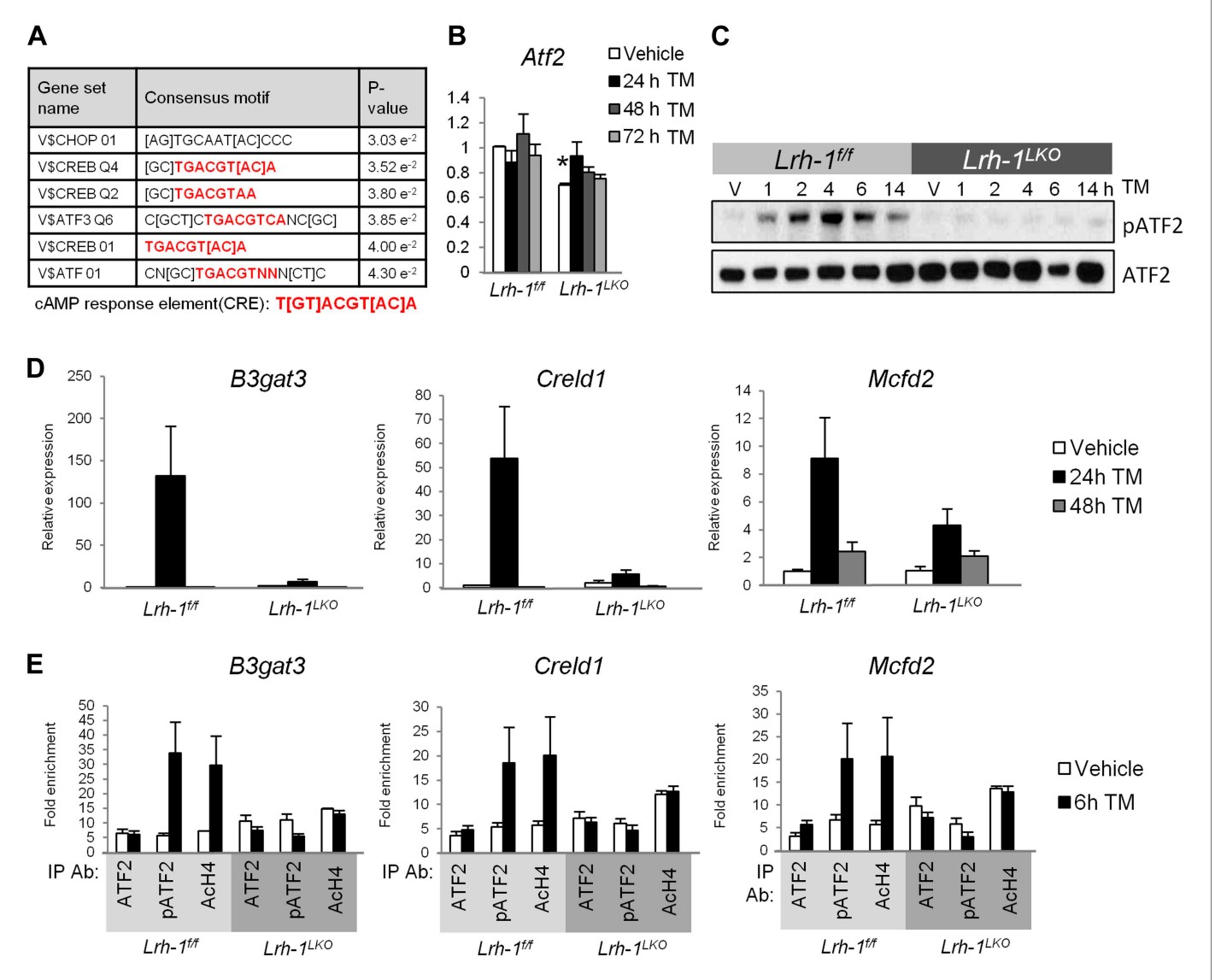

**Figure 4**. Microarray analysis suggests loss of ATF2 transcriptional ability in *Lrh-1* deficient mice. (**A**) Illumina Mouse Ref-8 arrays were performed for control and *Lrh-1LKO* mice treated with vehicle or tunicamycin (TM) for 24 hr (n = 3). Data were normalized and top 100 genes differentially induced by TM as assessed by fold change was analyzed by the Molecular Signatures Database software to identify overrepresented binding motifs. Transcription factors with binding sites significantly enriched in input gene promoters shown in chart. Motifs shown with significance at p<0.05. (**B**) Relative expression for *Atf2*. RNA was collected at designated timepoints for control and *Lrh-1LKO* mice treated with vehicle or 1 mg/kg TM (n = 3–6). Data were normalized to *Tbp* expression. Significance at p<0.01. (**C**) Immunoblot for total ATF2 and phospho-ATF2 (mT51/53; hT69/71), with pATF2 representing sites required to be phosphorylated for ATF2 activity. Primary hepatocytes were isolated from control and *Lrh-1LKO* mice and treated with 5 μg/ml TM or vehicle and protein collected at designated timepoints. Results representative of three independent experiments. (**D**) Relative expression for *B3gat3*, *Creld1*, and *Mcfd2*. RNA was collected at designated timepoints for control and *Lrh-1LKO* mice treated with vehicle or 1 mg/kg TM (n = 3–6). Data were normalized to *Tbp* expression. (**E**) Chromatin immunoprecipitation was performed from livers of control and *Lrh-1LKO* mice treated with vehicle or TM (1 mg/kg) for 6 hr (n = 4). Immunoprecipation was done with an anti-ATF2, pATF2 (69/71), or acetyl histone H4 antibody or rabbit IgG as a control. qPCR was used to determine binding by use of primers flanking binding site previously identified by ENCODE Projects.

between genotypes at 24 hr, indicating that the UPR branches are functional in *Plk3–/–* mice. As expected, wild-type mice had resolved ER stress by 72 hr, as evidenced by reduction of UPR transcription factor accumulation, but the sustained UPR signaling in *Plk3–/–* mice indicated that they were unable to resolve ER stress (*Figure 5J*), as also observed with chemical PLK3 inhibition. Thus, we conclude that

**Table 1.** Genes differentially induced by TM between control and Lrh-1-null mice and their regulation by ATF2 and LRH-1

| Gene | Known target of ATF2 | LRH-1 |
|---|---|---|
| PLK3 | ✓ | ✓ |
| GDF15 | | |
| FGF21 | * | |
| PMM1 | | |
| CRELD2 | | ✓ |
| NFIL3 | | |
| IGSF11 | | |
| GSTM2 | | |
| GOT1 | | |
| DPP9 | ✓ | ✓ |
| ST5 | | |
| CCBL1 | ✓ | ✓ |
| TMEM66 | ✓ | |
| KRTCAP2 | ✓ | ✓ |
| LRRC59 | ✓ | ✓ |
| CDK2AP2 | ✓ | ✓ |
| ST3GAL1 | | |
| TES | | ✓ |
| CCDC134 | ✓ | ✓ |
| UGT1L | | |
| ACOT2 | ✓ | |
| PLIN5 | | ✓ |
| LITAF | ✓ | |
| RPS13 | ✓ | |
| LGALS3BP | | |
| CRYM | | |
| SUPT5H | ✓ | |
| B3GAT3 | ✓ | |
| GRN | ✓ | |
| ARRDC4 | | |
| SLC30A7 | ✓ | |
| CRELD1 | ✓ | |
| NT5M | | |
| ARSG | | |
| SEP15 | | |
| MKNK1 | | ✓ |
| DDX52 | | |
| EXOSC5 | ✓ | |
| RABAC1 | ✓ | |
| RPS15 | ✓ | ✓ |
| D830014E11RIK | | |

*Table 1. Continued on next page*

loss of *Plk3* confers ER stress sensitivity similar to that observed in *Lrh-1^LKO* mice.

## Restoration of *Plk3* induction in *Lrh-1*-deficient mice rescues their ability to phosphorylate ATF2 and resolve ER stress

As loss of *Plk3* phenocopied the loss of *Lrh-1*, we predicted that restoration of stress-inducible *Plk3* induction in *Lrh-1^LKO* mice should rescue ATF2 phosphorylation and resolve ER stress. We generated a tetracycline-inducible adenovirus overexpressing mouse *Plk3* (Ad-Plk3), with *LacZ* used as a control (Ad-control). Primary hepatocytes from control and *Lrh-1^LKO* mice were transduced and later treated with vehicle or TM, as well as doxycycline to induce expression of *Plk3* or *LacZ*, and nuclear UPR accumulation of spliced XBP-1, ATF4, and cleaved ATF6 was assessed. Following TM treatment, control cells transduced with Ad-Plk3 or Ad-control showed comparable UPR responses at 24 hr, with resolution of ER stress by 48 hr. Treatment of *Lrh-1^LKO* cells with Ad-control resulted in no improvement in ER stress resolution capacity, as demonstrated by failure to resolve ER stress by 48 hr, similar to what was observed in *Lrh-1^LKO* mice treated with TM. In contrast, transduction of *Lrh-1^LKO* cells with Ad-Plk3 restored their ability to resolve ER stress, as evidenced by the absence of nuclear UPR transcription factor accumulation at 48 hr (**Figure 6A**). Thus, restoration of *Plk3* induction in *Lrh-1^LKO* cells is sufficient for the restoration of ER stress resolution.

To confirm that the restoration of *Plk3* induction in primary liver cells from *Lrh-1^LKO* mice could restore stress-inducible ATF2 phosphorylation, primary hepatocytes from *Lrh-1^LKO* mice were transduced and treated with TM as described above. In accordance with what we observed in non-transduced hepatocytes from *Lrh-1^LKO* mice, *Lrh-1^LKO* cells transduced with Ad-control were unable to phosphorylate ATF2 in response to stress (**Figure 6B**). However, when transduced with Ad-Plk3, cells from *Lrh-1^LKO* mice regained the ability to phosphorylate ATF2 in a stress-inducible manner, consistent with what we had previously observed in control mice (**Figure 4C**).

Lastly, we asked whether the striking phenotype of liver lipid accumulation observed in *Lrh-1^LKO* mice following ER stress could be resolved by restoration of *Plk3* induction. Primary hepatocytes from control and *Lrh-1^LKO* mice were isolated, transduced with adenovirus, and treated with TM as described above. Cells were stained with Lipidtox to visualize neutral lipids (red), and counterstained with DAPI. No differences in fat

*Table 1. Continued*

| Gene | Known target of | |
| --- | --- | --- |
| | ATF2 | LRH-1 |
| AVPI1 | ✓ | |
| RPS9 | ✓ | |
| CRAT | | ✓ |
| BHMT2 | | ✓ |
| B230217C12RIK | | |
| EIF3G | ✓ | ✓ |
| SLC25A28 | | ✓ |
| HR | | |
| CCL9 | | |
| ANXA4 | ✓ | ✓ |
| SMCO4 | | |
| SMOX | | |
| ARL14EP | ✓ | |
| SLC39A7 | | ✓ |
| ICA1 | | |
| ENTPD5 | | ✓ |
| PIWIL2 | | |
| ANG | ✓ | |
| MCFD2 | ✓ | ✓ |
| SOAT2 | | |
| SLC41A3 | ✓ | ✓ |
| MFGE8 | | |
| CYP4A14 | | |
| D12ERTD647E | | |

Microarray analysis was performed for control and *Lrh-1*[LKO] mice treated with vehicle or 1 mg/kg tunicamycin for 24 hr (n = 3). Genes were screened for those induced at least 1.5 fold by TM in control mice with significantly different induction in *Lrh-1*[LKO] mice by t-test (p<0.05). This list was filtered for those with differential expression between genotypes with TM treatment by t-test (p<0.05). Previously published genome-wide ATF2 and LRH-1 binding datasets were analyzed to identify genes with ATF2 or LRH-1 binding sites −500 to +275 bp from the TSS. Genes in our set were compared with these sets and genes that contain ATF2 or LRH-1 binding sites meeting the above criteria are marked.
*No binding site for ENCODE data set but a known ATF2 direct target (*Hondares et al., 2011*).

accumulation between genotypes or transduction with Ad-Plk3 or Ad-control were observed when cells were treated with vehicle or TM for 24 hr (*Figure 6C*). However, by 48 hr post stress, lipid staining was reduced in control cells, irrespective of transduction with either Ad-Plk3 or Ad-control. In *Lrh-1*[LKO] cells transduced with Ad-control, lipid accumulation was sustained at later timepoints, reminiscent of what we had observed in *Lrh-1*[LKO] mice treated with TM (*Figure 1A*). In contrast, *Lrh-1*[LKO] cells transduced with Ad-Plk3 exhibited lipid levels comparable to those in control cells. Since the overexpression of *Plk3* is quite minimal in our system (*Figure 6D*), we expect that conclusions drawn from these experiments are physiologically relevant. Overall, these studies demonstrate that *Plk3* expression is sufficient to rescue the defects in the *Lrh-1*[LKO] mice of ATF2 target gene induction and ER stress resolution in response to TM, as well as protect against metabolic derangement following stress.

## ATF2 mediates effects of PLK3 activation

Our bioinformatics analyses indicated that ATF2 is the predominant downstream effector of PLK3. However, PLK3 is known to phosphorylate other transcription factors, including c-Jun and p53 (*Table 2*). Further analysis of our gene set (*Table 1*) for overlap with known binding sites for these transcription factors in promoters of our genes indicated that, while c-Jun and p53 sites are enriched, ATF2 remains the most significant factor. To investigate the functional significance of ATF2 in ER stress resolution, we generated adenoviruses expressing mouse *Atf2* with mutations in essential phosphorylation sites (T51/53 to A51/53), similar to constructs previously reported (*Hayakawa et al., 2003*), to serve as a dominant negative (Ad-DN Atf2). We also generated adenovirus expressing a constitutively active *Atf2* (*Steinmuller and Thiel, 2003*) (Ad-C2/Atf2) or *LacZ* (Ad-control). Primary hepatocytes from control mice were transduced with Ad-control or Ad-DN Atf2 and treated with tunicamycin. We observed resolution of TM-induced ER stress by 48 hr when transduced with Ad-control, but sustained UPR activation when transduced with Ad-DN Atf2 (*Figure 6E*), even at moderate levels of viral overexpression (*Figure 6F*). We also prepared primary hepatocytes from control and *Lrh-1*[LKO] mice and transduced them with Ad-control or Ad-C2/Atf2. When transduced with Ad-control, control hepatocytes are able to resolve ER stress, whereas hepatocytes deficient in *Lrh-1* cannot (*Figure 6E*). However, transduction with Ad-C2/Atf2 increases the stress resolution capacity of *Lrh-1*[LKO] cells. Taken together, along with spliced *Xbp-1* expression quantifying these experiments (*Figure 6G*), we conclude that the loss of ATF2 activity sensitizes cells to ER stress, while overexpression of active ATF2 facilitates ER stress resolution.

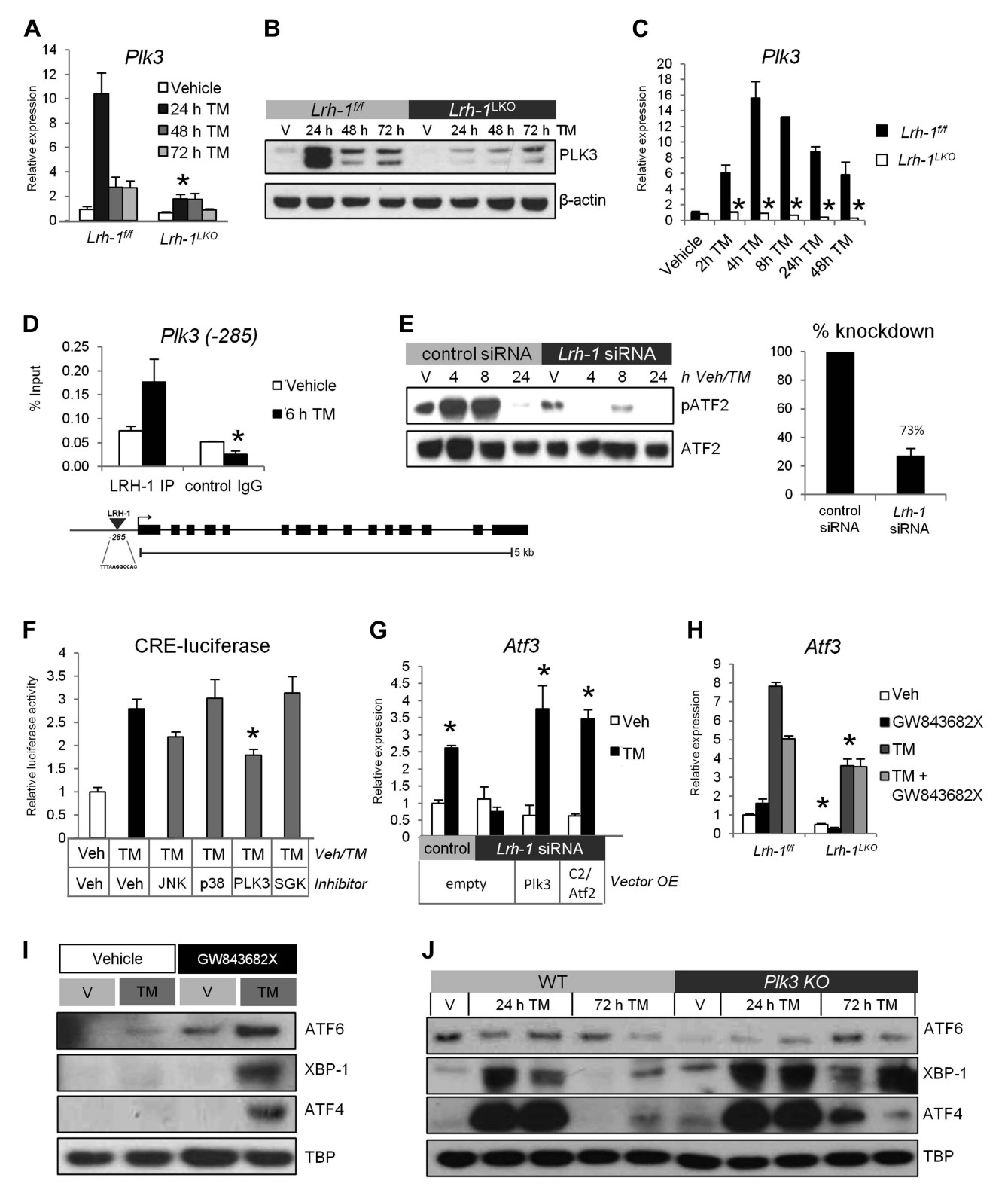

**Figure 5**. *Plk3* is a direct LRH-1 target that is required for induction of ATF2 target genes and ER stress resolution. (**A**) Quantitative PCR for *Plk3*. RNA was collected at designated timepoints for control and *Lrh-1*<sup>LKO</sup> mice treated with vehicle or 1 mg/kg tunicamycin (TM) (n = 3–6). Data were normalized to *Tbp* expression. Significance at p<0.01. (**B**) Immunoblot for PLK3 protein in control and *Lrh-1*<sup>LKO</sup> mice treated with vehicle or 1 mg/kg TM for designated
*Figure 5. Continued on next page*

Figure 5. Continued

timepoints. Liver tissue was boiled in Laemmli buffer to extract insoluble protein and samples were pooled (n = 2) prior to gel loading. β-actin was used as a loading control. (C) Quantitative PCR for *Plk3*. RNA was collected at designated timepoints for primary hepatocytes isolated from control and *Lrh-1^LKO* mice treated with vehicle or 0.5 µg/ml TM (n = 3). Data were normalized to *Tbp* expression. Significance at p<0.01. (D) A LRH-1 binding site in the *Plk3* promoter was identified 285 bases upstream of the TSS. Chromatin immunoprecipitation was performed from livers of control mice treated with vehicle or TM (1 mg/kg) for 6 hr (n = 3). Immunoprecipation was done with an anti-LRH-1 antibody or mouse IgG as a control. qPCR was used to determine binding by use of primers flanking binding site. Significance at p<0.05 between antibodies with error bars representing SEM. (E) TLR-3 cells were transfected with nonsilencing siRNA or siRNA against *Lrh-1* and nuclear protein was prepared at designated timepoints. Samples were immunoblotted for total ATF2 and phospho-ATF2 (mT51/53; hT69/T71) following treatment with vehicle or 1 ug/ml TM. Results representative of three independent experiments. (F) Relative luciferase activity for TLR-3 cells transfected with a cAMP response element (CRE)-luciferase reporter, Atf2, and Lrh-1. 48 hr after transfection, cells were treated with vehicle or 5 µg/ml TM and the following inhibitors: 10 µM D-JNKi for JNKs, 1 µM SB202190 for p38, 10 µM GW84362X for PLK1/PKL3, or 1 µM GSK650394A for SGK. 24 hr after treatment, cells were lysed, and luciferase activity was measured and normalized to β-galactosidase activity. Significance at p<0.01 as compared with TM treated cells (n = 3). Results representative of three independent experiments. (G) *Atf3* expression by qPCR in TLR-3 cells transfected with control or siRNA targeting *Lrh-1* (knockdown efficiency same as 5E), along with overexpression of constitutively active Atf2 (C2/Atf2), Plk3, or an empty vector. 48 hr post transfection, cells were treated with 1 µg/ml TM for 24 hr. Data were normalized to *Tbp* expression. Significance at p<0.01 for TM treated vs vehicle treated samples (n = 3). Results representative of three independent experiments. (H) *Atf3* expression by qPCR from primary hepatocytes from control and *Lrh-1^LKO* mice treated with vehicle or 5 µg/ml TM and 10 µM PLK3 inhibitor GW843682X for 24 hr. Data were normalized to *Tbp* expression. Significance at p<0.01 between genotypes (n = cells from 2–3 mice/group). (I) Wild-type mice were i.p. injected with PLK3 inhibitor GW843682X (1 mg/kg BW) or vehicle (DMSO). Mice were also i.p. injected with TM (1 mg/kg BW) or vehicle (DMSO). 48-hr post injection, liver tissue was collected and nuclear protein was isolated to assess accumulation of UPR transcription factors spliced XBP-1, cleaved ATF6, and ATF4. TBP was used as a loading control. Results representative of results from three mice. (J) Wild-type and *Plk3^−/−* mice were treated with TM (0.5 mg/kg) or vehicle. Nuclear UPR proteins were assessed by immunoblot at designated timepoints . TBP was used as a loading control. Results representative of four independent samples.

## Activation of LRH-1 induces *Plk3* and ATF2 target genes and increases resistance to ER stress

Based on the increased expression of *Plk3* and ATF2 target genes by tunicamycin with LRH-1 agonist treatment compared to TM alone (*Table 1*), we hypothesized that LRH-1 activation would promote increased ER stress resolution and cell survival. We treated primary hepatocytes from mice transgenic for human *Lrh-1* (h*Lrh-1 TG*) and null for mouse *Lrh-1* (m*Lrh-1 ^LKO*) with the non-lipid LRH-1 agonist RJW100 (*Whitby et al., 2011*). As predicted (*Table 1*), co-treatment with RJW100 and TM increased expression of *Plk3* above that by TM alone (*Figure 7A*), and this was dependent on h*Lrh-1* expression. The ATF2 target genes *Mcfd2* and *Atf3* could also be induced by RJW100 alone, which was also dependent on h*Lrh-1* (*Figure 7B*). We observed no effect of RJW100 on induction of target genes dependent on any UPR transcription factor (*Figure 7C*), leading us to conclude that the effect of RJW100 is specific to the ER stress responsive pathway initiated by LRH-1. We then treated h*Lrh-1 TG*; m*Lrh-1 ^LKO* mouse primary hepatocytes with a low or high dose of TM, along with RJW100. For both low and high doses of TM, co-treatment with RJW100 increased ability of cells to resolve ER stress, as evidenced by less sustained UPR signaling by 48 hr post treatment (*Figure 7D*). To evaluate cell survival, we treated primary hepatocytes from h*Lrh-1 TG*; m*Lrh-1 ^LKO* and m*Lrh-1 ^LKO* mice with a range of TM doses, along with RJW100. RJW100 decreased cell death in h*Lrh-1 TG*; m*Lrh-1 ^LKO* cells back to untreated levels for all but the highest TM dose, with significant reduction even at this dose (*Figure 7E*). Mice lacking m*Lrh-1* were more sensitive to TM at all doses and RJW100 had no beneficial effects on cell survival (*Figure 7E*). Taken together, our results indicate that h*Lrh-1* is able to compensate for loss of m*Lrh-1* in ER stress resolution, and that LRH-1 agonism potently increases the ability of cells to resolve high levels of ER stress.

## Discussion

Our studies have identified an unexpected yet essential pathway for hepatic ER stress resolution initiated by the nuclear receptor LRH-1 (*Figure 8*). Following ER stress, LRH-1 is recruited to the *Plk3* promoter and dramatically induces transcription of this atypical kinase. This is essential for ATF2 phosphorylation, which is profoundly deficient in *Lrh-1^LKO* hepatocytes in response to stress. PLK3 is required for ATF2 activation following ER stress, and that *Plk3^−/−* mice, like *Lrh-1^LKO* mice, are defective in ER stress resolution. The similar impact of loss of *Lrh-1* and loss of *Plk3* predicted that restoration of *Plk3* induction to *Lrh-1^LKO* mice could be sufficient to rescue their ability to resolve ER stress. We found that restoring *Plk3* induction to primary *Lrh-1^LKO* hepatocytes did restore their ability to phosphorylate ATF2

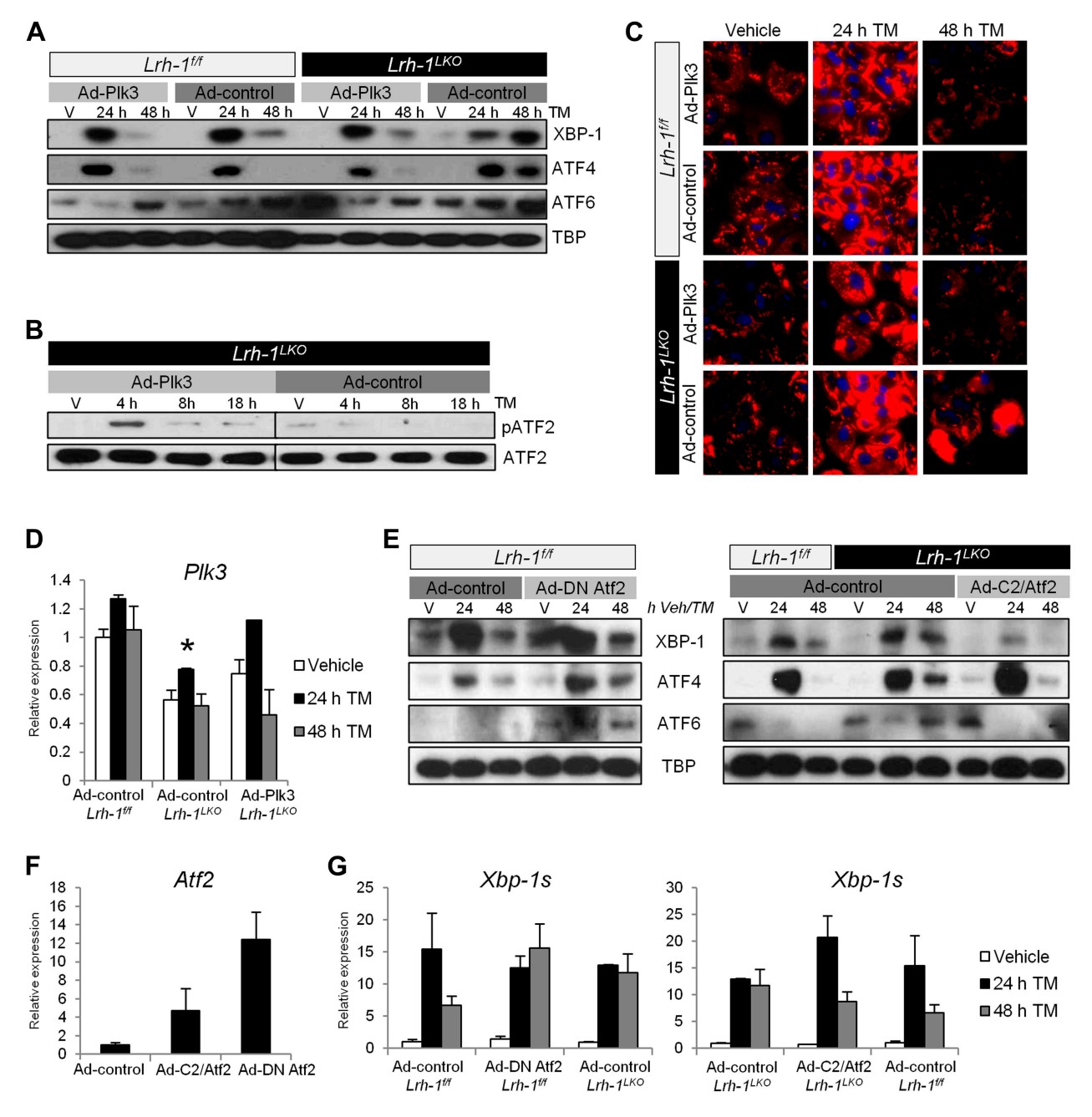

**Figure 6**. Restoration of Plk3 induction rescues ATF2 phosphorylation and ER stress resolution in *Lrh-1^LKO* mice, and loss or gain of ATF2 transcriptional activity also alters ER stress resolution capacity. (**A**) Primary hepatocytes were prepared from *Lrh-1^f/f and Lrh-1^LKO* mice and transduced with Ad-Plk3 or Ad-control at a MOI of 100. Cells were treated 36 hr later with vehicle or tunicamycin (TM) (0.01 µg/ml) and doxycycline (1 µg/ml) to induce Plk3 or LacZ control. Nuclear protein was obtained at timepoints indicated and immunoblotted for UPR transcription factors, with TBP as a loading control. Results are representative of three independent experiments. (**B**) Primary hepatocytes were prepared from *Lrh-1^LKO* mice and transduced with Ad-Plk3 or Ad-control at a MOI of 100. Cells were treated 36 hr later with vehicle or TM (1 µg/ml) and doxycycline (1 µg/ml) to induce Plk3 or LacZ control. Nuclear protein was collected at indicated timepoints and immunoblotted for pATF2 (69/71) and total ATF2. Samples are from same gel and brightness/contrast was adjusted equally prior to cropping. Results are representative of three independent experiments. (**C**) Primary hepatocytes were prepared from *Lrh-1^f/f* and *Lrh-1^LKO* mice and transduced with Ad-Plk3 or Ad-control at a MOI of 100. Cells were treated 36 hr later with vehicle or TM (0.01 µg/ml) and doxycycline (1 µg/ml) to induce Plk3 or LacZ control. Cells were fixed at various timepoints and stained for lipids (red) using Lipidtox dye and counterstained with DAPI. Magnification is 40X (objective). Results are representative of three independent experiments. (**D**) Relative expression of *Plk3* by qPCR. Primary hepatocytes from control and *Lrh-1^LKO* mice were transduced with Ad-Plk3 or Ad-control at a MOI of 100 and treated with 1 µg/ml doxycycline

*Figure 6. Continued on next page*

*Figure 6. Continued*

and vehicle or 0.01 μg/ml TM. Data were normalized to *Tbp* expression. Significance at p<0.01 between genotypes (n = cells from 3–4 mice). (**E**) Primary hepatocytes from control mice were transduced with Ad-DN Atf2 (Atf2 T51A/T53A to serve as a dominant negative) or Ad-control at a MOI of 100 and treated with 5 ng/ml TM. Primary hepatocytes from control and *Lrh-1^LKO* mice were transduced with Ad-C2/Atf2 (expressing a constitutively active Atf2) or Ad-control at a MOI of 100 and treated with 5 ng/ml TM. Nuclear protein was obtained at timepoints indicated and immunoblotted for UPR transcription factors, with TBP as a loading control. Results are representative of samples from three mice. (**F**) *Atf2* expression by qPCR to quantify viral *Atf2* overexpression. Primers were chosen to amplify regions identical between wildtype Atf2, C2/Atf2, and DN Atf2. Primary hepatocytes from control mice (n = 3–4) were transduced at a MOI of 100 and treated with vehicle for 24 hr prior to RNA collection. (**G**) Spliced *Xbp-1* expression by qPCR for primary hepatocytes from control and *Lrh-1^LKO* mice (n = 3–4) transduced with Ad-control, Ad-DN Atf2, or Ad-C2/Atf2 and treated with vehicle or 5 ng/μl TM. Data were normalized to Tbp expression.

and also resulted in decreased fat accumulation at later times after ER stress. More importantly, the *Lrh-1^LKO* hepatocytes resolved ER stress similar to wild-type cells when *Plk3* induction was reinstated.

This led us to consider whether activation of ATF2 is also required for ER stress resolution. Since functional redundancy with the transcription factors Atf7 and Cre-bpa (Creb5) complicates study of ATF2 function in mice (***Breitwieser et al., 2007***), we overexpressed a dominant negative ATF2 in hepatocytes. Transduction of wild-type hepatocytes with this construct diminishes ability to resolve ER stress, consistent with that observed following the loss of *Lrh-1* or *Plk3*. We also found that overexpression of a constitutively active ATF2 resulted in improved ability to resolve ER stress. These results indicate that ATF2 activation is sufficient for ER stress resolution, and that activation of ATF2 or its heterodimeric partners, including ATF7 and/or CRE-BPa, is necessary. Determining genome-wide binding of ATF2 and other PLK3-responsive transcription factors in the context of ER stress should enhance our understanding of the downstream targets of this essential pathway.

The induction of LRH-1 target genes following ER stress and its increased recruitment to the *Plk3* promoter suggests that LRH-1 is transcriptionally activated in response to TM treatment. In general, this could result from either signaling pathways that phosphorylate, or otherwise activate LRH-1 or its coregulators, or to increased production of an endogenous LRH-1 agonist. It could also result from increased expression of *Lrh-1* following ER stress. Unfortunately, both the very limited information on specific signaling pathways that promote LRH-1 transactivation and the absence of information on

**Table 2.** Enrichment of known transcription factor binding sites in our gene set (***Table 1***) of differentially regulated genes by ER stress between control and *Lrh-1LKO* mice

| Transcription factor name: | Known phosphorylation by PLK3: | Overlap of target genes with our gene set (*Table 1*): | Dataset used for analysis: |
|---|---|---|---|
| ATF2 | T71 (***Wang et al., 2011b***) | 3.83E-08 | ATF2 binding in G12878 cells (ENCODE EH002306) |
| Lrh-1 | none | 1.40E-04 | Lrh-1 binding in mouse liver (***Chong et al., 2012***) |
| p53 | S20 (***Xie et al., 2001***) | 1.37E-04 | p53 binding in U2OS cells treated with Nutlin-3 (***Menendez et al., 2013***), which results in S20 phosphorylation (***Valentine et al., 2011***) |
| c-Jun | S63 and S73 (***Wang et al., 2011a***) | 2.95E-02 | c-Jun binding in CH12 cells (ENCODE EM001943), in which S63 may be constitutively phosphorylated like other B-lymphoma lines (***Gururajan et al., 2005***) |
| NRF2 | None | NS | NRF2 binding in lymphoid cell lines treated with sulforaphane (***Chorley et al., 2012***) |

Transcription factors known to be phosphorylated by PLK3 (ATF2, p53, and c-Jun) are summarized here, along with NRF2 to represent the oxidative stress response and LRH-1. Overlap of known transcription factor binding sites −500 to +b250 bp of the TSS for genes in Table 1 was calculated, and significance was determined using hypergeometric distribution tests.

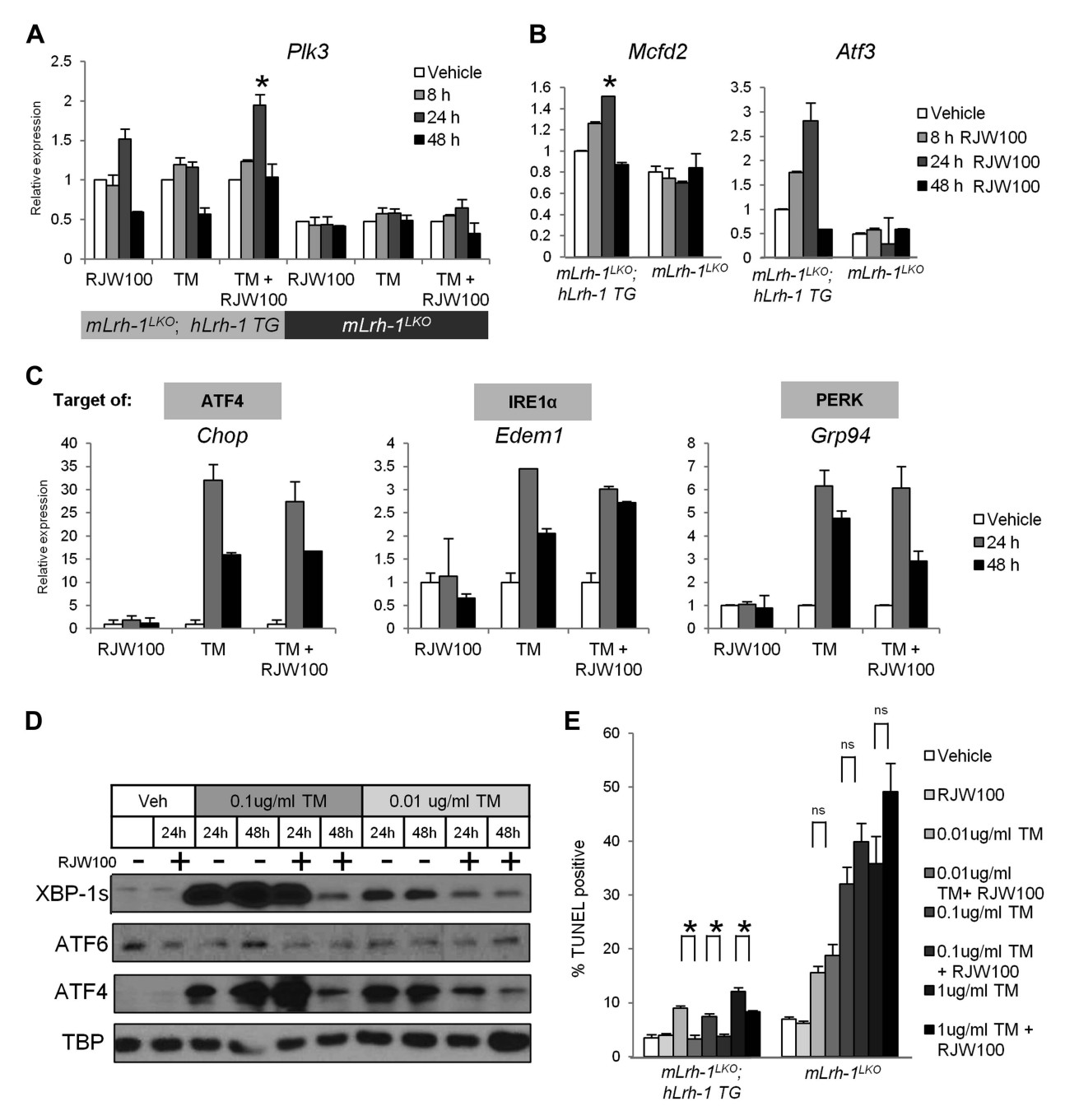

**Figure 7**. LRH-1 agonism promotes *Plk3* and ATF2 target gene expression and increases resistance to ER stress independent of the UPR. (**A**) Quantitative PCR for *Plk3*. RNA was collected at designated timepoints for primary hepatocytes isolated from h*Lrh-1 TG*; m*Lrh-1 LKO* and m*Lrh-1LKO* mice treated with 10 uM RJW100 and/or 0.01 ug/ml TM (n = cells from 3 mice). Data were normalized to *Tbp* expression. Significance at p<0.05 as compared with vehicle treatment. (**B**) Quantitative PCR for *Mcfd2* and *Atf3*. RNA was collected at designated timepoints for primary hepatocytes isolated from h*Lrh-1 TG*; m*Lrh-1 LKO* and m*Lrh-1LKO* mice treated with 10 µM RJW100 (n = cells from 3 mice). Data were normalized to *Tbp* expression. Significance at p<0.05 as compared with vehicle treatment. (**C**) Primary hepatocytes were isolated from h*Lrh-1 TG*; m*Lrh-1 LKO* mice (n = cells from 3 mice) and treated with 10 µM RJW100 and/or 0.01 µg/ml TM. RNA was collected at designated timepoints and qPCR performed for UPR target genes. Data were normalized to *Tbp* expression. No significance between groups. (**D**) Primary hepatocytes were isolated from h*Lrh-1 TG*; m*Lrh-1 LKO* mice and treated with 10 µM RJW100 and/or 0.01 µg/ml or 0.1 µg/ml TM. Nuclear protein was obtained at timepoints indicated and immunoblotted for UPR transcription factors, with TBP as a loading control. Results are representative of three independent experiments. (**E**) Primary hepatocytes were isolated from h*Lrh-1 TG*; m*Lrh-1 LKO* and m*Lrh-1LKO* mice (n = cells from three mice) and treated with 10 µM RJW100 and/or 0.01–1 µg/ml TM. Cells were fixed 48 hr post treatment and TUNEL staining was performed and quantified. Ratio of TUNEL-positive to DAPI-positive cells was calculated from four fields of each slide (n = 3). Significance at p<0.01.

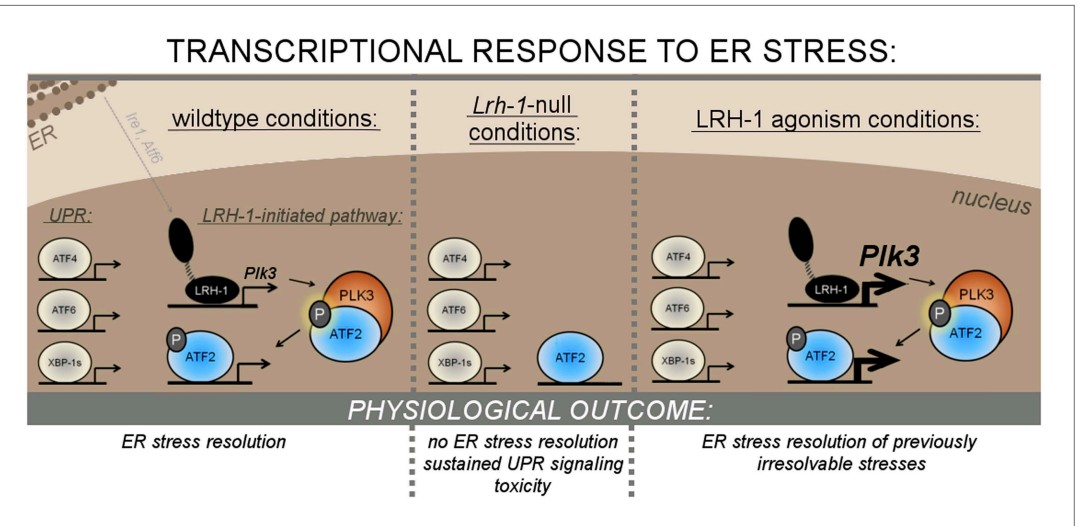

**Figure 8**. Mechanism of LRH-1's requirement in ER stress resolution.

endogenous LRH-1 ligands severely limits approaches to determine whether LRH-1 is post-translationally regulated following ER stress. However, we observed that the stress-dependent induction of *Lrh-1* expression is dependent on the UPR components IRE1a and ATF6, and IRE1a also appears to impact LRH-1 transactivation. It is possible that transcriptional regulation of LRH-1 following ER stress is at least partially under control of the UPR in this regard. However, it is quite unlikely that IRE1a directly regulates LRH-1 protein, based on the constitutive nuclear localization of LRH-1 following ER stress. Future studies will be needed to characterize post-translational control of LRH-1 in a variety of contexts, including ER stress.

More broadly, the ligand responsiveness of LRH-1 suggests that it could become a therapeutic target in human disorders associated with chronic ER stress. Our results suggest that activation of LRH-1 by RJW100 heightens induction of *Plk3* and ATF2 target genes. Additionally, treatment of cells transgenic for human *Lrh-1* with RJW100 results in the ability to resolve high levels of ER stress and protection from cell death following stress. Therefore, targeting LRH-1 in vivo with agonists or newly developed antagonists (*Rey et al., 2012*) may allow selective augmentation or inhibition of ER stress resolution, which could prove beneficial in numerous human disorders. This is of particular interest since other strategies for targeting the UPR have proven to be difficult, and no chemical compounds have been successfully used in mice that directly activate downstream UPR components.

Although it has not been previously linked to the UPR, ATF2 has been associated with other cellular stress responses, including those that mitigate DNA damage (*van Dam et al., 1995*) and oxidative stress (*Kurata, 2000*), with these roles dependent on ATF2 phosphorylation. This raises the interesting question of whether this LRH-1 initiated pathway contributes to a broader, kinase-driven cell defense system that can be activated by diverse stresses. Highly conserved, kinase-driven pathways are increasingly being recognized for not only being activated by ER stress, but also for facilitating ER stress resolution. In mammalian cells, specific MAP kinases have been identified that are activated by ER stress and diminish cell death following stress (*Hu et al., 2004*). A screen for genes whose loss confers sensitivity to ER stress in *S. cerevisae* also uncovered twelve MAP kinase components, remarkable in scale considering that the yeast UPR only contains a single branch consisting of *IRE1* and substrate *HAC1* (*Chen et al., 2005*). We provide an additional example of such a kinase-driven ER stress resolution pathway, in which LRH-1 protects the cell through *Plk3* induction and subsequence ATF2 phosphorylation. While PLK3 is not a MAP kinase, it is important to note that ATF2 is otherwise phosphorylated by MAP kinases p38 and JNK. This suggests that the presence of LRH-1 adds an additional layer of control onto these MAP kinase pathways through activation of the atypical kinase. We are currently investigating the conservation of this response utilizing simpler model organisms, as well as investigating whether LRH-1 and downstream factors PLK3 and ATF2 are required for resolution of broader cell stresses.

Overall, we conclude that this novel LRH-1-*Plk3* pathway represents a core pathway that is required for hepatic ER stress resolution. Although it is as essential as any of the three canonical arms of the UPR, and appears to function independently of them, it cannot be considered a fourth arm because there is no evidence that it directly contributes to restoration of ER function, for example by promoting protein folding, and also because LRH-1 is not universally expressed. Nonetheless, our results open new directions for therapeutic targeting of ER stress in human disease, and suggest that this pathway, which links highly conserved kinase signaling and resolution of ER stress, is a prime example of the emerging integration of broad cell stress responses and regulation of protein homeostasis.

## Materials and methods

### Animal studies

*Lrh-1* liver specific knockout (*Lrh-1^LKO^*) mice were obtained by crossing mice with an *Lrh-1* allele flanked by LoxP sites (*Lrh-1^f/f^*) with *albumin-Cre* transgenic mice. *Lrh-1^f/f^* mice were provided by the Kliewer/Mangelsdorf lab and have been previously described (*Lee et al., 2008*); *albumin-Cre* transgenic mice were provided by Bert O'Malley's laboratory at Baylor College of Medicine. Wild-type and *Plk3−/−* mice were obtained from Peter Stambrook's lab and have been previously described (*Myer et al., 2011*). Male mice 8–12 weeks of age were injected intraperitoneally with tunicamycin (1 mg/kg body weight) or vehicle (2% DMSO) in 150 mM dextrose. A dose of 0.5 mg/kg TM was used for *Plk3* WT and *Plk3−/−* mice due to sensitivity of this strain to the drug. Conditionally expressed humanized *Lrh-1* transgenic (*hLrh-1* TG) mice were a kind gift from Franco DeMayo and were generated as previously described for *hCoup-TF1* (*Wu et al., 2010*) unless otherwise indicated. In short, cDNA for human *Lrh-1* was cloned into a shuttle vector, yielding an N terminus-FLAG-myc tandem-tagged protein. The shuttle vector was homologously recombined with a base vector containing two ROSA26 genomic sequences and a loxP-STOP-loxP cassette and transformed into 294-Flp cells for generation of the targeting construct. The targeting construct was linearized and electroporated into AB ES cells, instead of R1 cells as previously described. ES clones were screened and C57 chimeras were produced as previously described. Mice were maintained in a mixed background and crossed with *Lrh-1^f/f^* mice with one allele of *albumin-Cre*. We confirmed that *hLrh-1* TG; *Lrh-1^LKO^* mice (*hLrh-1* TG/+; Lrh-1^f/f^; *albumin-Cre*/+) had robust expression of *hLrh-1* by PCR. Methods were approved by Baylor College of Medicine's Institutional Animal Care and Use Committee.

### Protein isolation

Nuclear and cytoplasmic fractions were obtained from fresh liver tissue or primary hepatocytes. Fractionation was adapted from a protocol written for cultured cells obtained from the lab website of David Ron (http://ron.medschl.cam.ac.uk/protocols/NucCyto.html). Briefly, tissue was dounce homogenized in a solution containing 10 mM HEPES, 50 mM NaCl, 0.5M sucrose, 0.1 mM EDTA, and 0.5% Triton X100. This suspension was spun at 1000 rpm at 4°C to pellet nuclei. Primary hepatocytes were simply harvested in the above solution and spun as above. The cytoplasmic supernatant was re-spun and resulting supernatant was collected. Nuclei were washed in a solution containing 10 mM HEPES, 10 mM KCl, 0.1 mM EDTA, and 0.1 mM EGTA and pellets were then suspended in a solution of 10 mM HEPES, 500 mM NaCl, 0.1 mM EDTA, 0.1 mM EGTA, and 0.1% NP-40. This suspension was vortexed and spun to yield a supernatant containing nuclear proteins. To obtain insoluble proteins (containing PLK3), we homogenized liver tissue in Laemmli buffer with B-mercaptoethanol and boiled for 10 min.

### Immunoblotting

Samples were separated on 4–12% Bis-Tris gels (Invitrogen, Carlsbad, CA). For cytoplasmic fractions or whole cell lysates, 100 µg of total protein was loaded. For nuclear fractions, 10 µg of total protein was loaded. The following antibodies and conditions were used: cleaved PARP (9544; 1:500; Cell Signaling, Danvers, MA), caspase-6 (9762; 1:500; Cell Signaling), cleaved caspase-3 (9664; 1:500; Cell Signaling), XBP-1 (sc-7160; 1:200; Santa Cruz, Dallas, TX), ATF4 (sc-200; 1:200; Santa Cruz), ATF6 (IMG-273; 1:200; Imgenex, San Diego, CA); LRH-1 (PP-H2325-10; 1:200; Perseus Proteomics, Tokyo, Japan), pATF2 (9225S; 1:500; Cell Signaling [note: we have experienced difficulties with recent batches]), ATF2 (ab47476; 1:1000; Abcam, Cambridge, England); and PLK3 (4896S; 1:400; Cell Signaling). Immobilon Western substrate (Millipore, Billerica, MA) was used for detection.

## Quantitative PCR

RNA was isolated from snap-frozen liver tissue by homogenization in TriZol reagent (Invitrogen) and precipitation with ethanol. cDNA was synthesized from 1 µg RNA using Qiagen's QuantiTect reverse transcription kit. Quantitative PCR was run on a Roche LightCycler 480 with Perfecta SYBR Green FastMix obtained from Quanta BioSciences (Gaithersburg, MD). Samples were run in triplicate from 3 to 6 samples/group and expression was normalized to *Tbp*. Standard curves were ran for each primer set and relative fold changes were calculated with the ΔΔCt method. Primer sequences are available in *Supplementary file 1*.

## Primary hepatocyte isolation

Mice were anesthetized with a lethal dose of tribromoethanol and perfused with Earle's balanced salt solution (EBSS) containing 5 mM EGTA. Perfusion was accomplished by inserting a 25 G needle into the inferior vena cava, cutting the portal vein, and using a peristaltic pump to deliver perfusion solutions. Following perfusion with EBSS, Hank's balanced salt solution (HBSS) with 100 U/ml collagenase and containing trypsin inhibitor was perfused. The liver was then removed and massaged to obtain dissociated cells in hepatocyte wash media (17704-024; Invitrogen). The cells were passed through a mesh to obtain a single cell suspension. This suspension was layered onto Percoll (P4937; Sigma, St. Louis, MO) and spun at 600 rpm for 10 min to isolate dead cells. Live cells pelleted were washed and viability was assessed. The cells were plated in collagen-coated dishes in Williams E media (12551; Invitrogen) containing insulin-transferrin-selenium supplementation. In studies in which cells were not transduced with adenovirus, cells were treated 12 hr later. In studies in which cells that were transduced with adenovirus, adenovirus was added 6 hr post plating at a multiplicity of infection of 100. The cells were kept for 36 hr in the same media prior to treatment.

## Generation of inducible *LacZ* and *Plk3*-expressing adenoviruses

We used Clontech's Adeno-X Adenoviral System 3 to generate tetracycline-inducible adenovirus according to manufacturer's instructions. We cloned the coding sequence for mouse *Plk3* from cDNA, along with supplied *LacZ* control fragment, into the pAdenoX-Tet3G vector. Plasmids were linearized with PacI and transfected into 293T/17 cells with Lipofectamine 2000 (Invitrogen). Titre was determined by use of Adeno-X Rapid Titer Kit (Clontech, Mountain View, CA). Virus was delivered experimentally as crude lysate.

## Generation of *LacZ*, DN *Atf2*, and C2/*Atf2*-expressing adenoviruses

We used the modified AdEasy system (*Luo et al., 2007*) to generate non-inducible adenoviruses expressing *LacZ*, DN *Atf2*, and C2/*Atf2*. To construct Ad-DN Atf2, we cloned the coding sequence for mouse Atf2 (NM_001025092) and used overlap extension PCR to introduce threonine to alanine mutations (T51A; T53A) corresponding to amino acids T69/71 in human ATF2. To construct Ad-C2/Atf2, we cloned C2/Atf2 from the pCMV-Flag-C2/Atf2 vector (*Steinmuller and Thiel, 2003*) generously provided by Dr Gerald Thiel. As previously described, this vector contains the DNA-binding domain of ATF2 fused with the activation domain of ATF4 to generate a constitutively active ATF2 mutant. These sequences, along with the coding sequence for LacZ, were digested with XhoI and KpnI and ligated into pShuttle-CMV (Addgene, Cambridge, MA). These vectors were electroporated into BJ5183-AD-1 cells (Stratagene, La Jolla, CA), which contain AdEasy-1 encoding the adenoviral backbone, for recombination. Potential recombinants were screened and successful recombinants were PacI linearized and transfected in 293 cells by calcium phosphate transfection. Titre was determined by use of Adeno-X Rapid Titer Kit (Clontech). Virus was delivered experimentally as crude lysate.

## Thioflavin T staining

Primary hepatocytes were plated on collagen-coated glass coverslips and treated with TM. The cells were washed and fixed in 4% paraformaldehyde for 20 min at room temperature. The cells were again washed and stained with 500 µM Thioflavin T in PBS for 3 min at room temperature. Fluorescence was visualized and imaged using a Zeiss Axioplan 2 microscope (GFP filter).

## TLR-3 cell culture

TLR-3 cells were obtained from the Health Science Research Resources Bank (HSRRB), part of the Japan Health Sciences Foundation. These cells were derived from C57Bl/6 mouse hepatocytes immortalized with a temperature sensitive large T antigen (tsSV40 large T antigen). We cultured cells at the

permissive temperature of 33°C on collagen-coated plates. Cells were cultured in DMEM supplemented with 5% FBS, 1x insulin-transferrin-selenium supplement (Invitrogen), and 10 ng/ml EGF.

## GFP-LRH-1 imaging

TLR-3 cells were grown on collagen-coated plates and transfected with N-terminal tagged EGFP-LRH-1 (*Atanasov et al., 2008*), a kind gift of Dr Thomas Brunner. The cells were treated with TM and fixed with 4% paraformaldehyde for 20 min at room temperature. The cells were counterstained with DAPI and visualized with a Zeiss Axioplan 2 microscope (GFP and DAPI filters).

## siRNA knockdown of Lrh-1 and UPR components

For primary hepatocyte experiments, we plated cells at half the typical density (31.19 thousand per 24-well plate well). siRNAs were purchased from Invitrogen (Stealth siRNA) and validated with the exception of siRNA to *Ire1a*, which was purchased as a pool from Dharmacon (Lafayette, CO) (ON-TARGETplus SMARTpool). 50 pmol siRNA was transfected into each well using Lipofectamine RNAiMAX transfection reagent (Invitrogen). Cells were treated with TM 48 hr post transfection and knockdown validated at 56 hr (representing the latest timepoint collected). For experiments in TLR-3 cells, cells were plated at 70% confluency in collagen-coated plates. *Lrh-1* siRNA was purchased from Invitrogen (Stealth siRNA) and 200 pmol was transfected in each well of a 12 well plate. Cells were treated with TM 48 hr post transfection.

## Luciferase assay

TLR-3 cells were plated in collagen-coated 24-well plates and transfected with 200 ng cAMP reponse element-luciferase (CRE-luciferase) reporter (Qiagen, Valencia, CA), 150 ng β-galactosidase, 200 ng mouse LRH-1, and 200 ng ATF2 (Addgene). 48 hr later, the cells were treated with TM or the following kinase inhibitors: 10 µM D-JNKi for JNKs (Sigma), 1 µM SB202190 for p38 (Tocris), 10 µM GW84362X for PLK1/PKL3 (Tocris, Bristol, UK), or 1 µM GSK650394A for SGK (Tocris). 24 hr after treatment, cells were lysed in Tropix lysis buffer (100 mM potassium phosphate, 0.2% TritonX-100, pH 7.8) plus DTT. Lysates were plated in triplicates in 96-well plates and 85 µl of reaction buffer was automatically injected by the luminometer. Reaction buffer for each well was prepared as follows: 0.7 µl galacton (Applied Biosystems, Foster City, CA), 88 µM luciferin, 2.4 mM ATP, and 11.9 mM $MgCl_2$ in 0.11M Tris-phosphate buffer, pH 7.8. Firefly luciferase activity was measured and samples were incubated 1 hr at room temperature. 100 µl Tropix Accelerator (Applied Biosystems) was then automatically injected and measured to quantify β-galactosidase activity, which was used to normalize luciferase activity values.

## Isolation and determination of liver triglycerides and non-esterified fatty acids

Lipid was extracted from liver by homogenizing in 9 volumes PBS and was added to a chloroform:methanol (2:1) mixture. PBS was added and the samples were spun at 3000 rpm for 10 min at 4°C. The bottom layer was removed and solute evaporated overnight. Lipid was resuspended in 1% Triton-X in ethanol for 4 hr with rotation. This was then used to calculate liver triglycerides and non-esterified fatty acids using Thermo Scientific's (Waltham, MA) Infinity Triglyceride kit and Wako's (Richmond, VA) NEFA kit, respectively. Triglyceride was extracted from primary hepatocytes by scraping cells into PBS and shaking at 2500 rpm for 15 s (2x) with a MagNA Lyser bead homogenizer (Roche, Basel, Switzerland). The samples were sonicated for 10 s prior to spinning at 4000 rpm 5 min to clear sample. Triglycerides were quantified with Thermo Scientific's Infinity Triglyceride kit and protein quantified by Bradford assay.

## Oil Red O staining

Liver tissue was frozen in optimum cutting temperature compound (Sakura Finetek, Torrance, CA) on dry ice. Tissue was finely cut using a cryostat, fixed, and stained with Oil Red O solution. Nuclei were also counterstained with hematoxylin. Tissue cutting and staining was performed by the Comparative Pathology Laboratory at Baylor College of Medicine.

## TUNEL staining

Liver tissue was frozen in optimum cutting temperature compound (Sakura Finetek) on dry ice. Tissue was finely cut using a cryostat by the Comparative Pathology Laboratory at Baylor College of Medicine. We then performed TUNEL staining in our lab with use of Takara's In Situ Apoptosis Detection Kit (Otsu, Japan). Briefly, tissue was fixed in acetone for 30 min and washed. Tissue was permeablized with buffer contained in kit for 5 min and incubated with labeling mix (containing TdT enzyme and

fluorescein-dUTP) for 90 min at 37°C. Tissue was then washed and mounted with a medium containing DAPI. Fluorescent images were captured using a Zeiss Axioplan 2 microscope at 400X using OpenLab 3.1.5 software. For primary hepatocytes, the cells were fixed in 4% PFA for 1 hr and permeabilized (0.1% Triton-X100 in 0.1% sodium citrate) for 5 min. The cells were washed and stained with TUNEL reaction mixture (2.5 mM cobalt chloride, 0.4 U/µl Tdt enzyme, and 2 µM fluorescein-dUTP) for 60 min at 37°C. The cells were washed and stained with DAPI and counted with a Zeiss Axioplan 2 microscope.

## Partial hepatectomy

Surgical removal of 2/3 liver mass was achieved by anesthetizing mice with tribromoethanol and resecting the three most anterior lobes of the liver. This was accomplished by tying off the lobes with silk suture to stop blood flow and cutting immediately below the suture. The right and left medial lobe were sutured together and the left lateral lobe was sutured separately. Mice undergoing sham surgery underwent anesthesia and laparotomy but no liver lobes were removed.

## Microarray analysis

Mice were treated with TM (1 mg/kg body weight) or vehicle for 24 hr. Total RNA from liver tissue was extracted using Trizol reagent (Invitrogen) and purity of the RNA was assessed by Agilent 2100 Bioanalyzer. 500 ng of RNA was reverse transcribed into cRNA and biotin-UTP labeled using the Illumina TotalPrep RNA Amplification Kit (Ambion, Austin, TX). cRNA was quantified using an Agilent Bioanalyzer 2100 and hybridized to the Illumina mouseRefseq-8v2 Expression BeadChip using standard protocols (Illumina, San Diego, CA). Image data was converted into unnormalized Sample Probe Profiles using the Illumina BeadStudio software. Arrays were normalized with Chipster software (*Kallio et al., 2011*) using default settings for Illumina arrays. To identify over-represented transcription factor binding sites, we used the Molecular Signatures Database, part of the Gene Set Enrichment Analysis software (*Subramanian et al., 2005*). We analyzed the top 100 genes induced at least 1.5-fold by tunicamycin in control mice with significantly different induction in *Lrh-1^LKO^* mice. Promoters from the input gene list were analyzed for enriched presence of 615 different known transcription factor binding sites (as defined in the TRANSFAC version 7.4 database). p<0.05 was considered significant. To generate a table of genes with significantly different induction by TM between genotypes, we filtered for genes induced at least 1.5-fold by TM in control mice with significantly different induction in *Lrh-1^LKO^* mice by t-test (p<0.05). We secondarily filtered for those with significantly different expression when treated with TM by t-test (p<0.05). For genome-wide binding of ATF2, we downloaded ENCODE data (wgEncodeEH002306) as a BED file and used HOMER software (*Heinz et al., 2010*) to annotate distance to TSS for each set of peak coordinates. For genome-wide binding of LRH-1, we obtained a BED file from the lab of Tim Osborne (*Chong et al., 2012*) and annotated distance to TSS with HOMER. For DLPC and TM-treated samples, control mice were treated with 100 mg/kg DLPC by oral gavage as previously described (*Lee et al., 2011*) every 12 hr (twice before TM and twice afterwards) and 1 mg/kg TM by i.p. injection. Microarray for these samples was performed alongside previously described samples. Fold change was calculated for TM-treated samples and compared to TM- and DLPC-treated samples (n = 3).

## Lipid staining of primary hepatocytes

Primary hepatocytes were washed and subsequently fixed in fresh 4% paraformaldehyde for 30 min. Following this, the cells were washed and then stained with Deep Neutral Red Lipidtox (Invitrogen) for 1 hr. The cells were then counterstained with DAPI and imaged with a Zeiss Axioplan 2 microscope at 400X using OpenLab 3.1.5 software. Merged images were enhanced for contrast using identical settings with GNU Image Manipulation Program.

## Statistical analysis

2-way ANOVA with Bonferroni post-hoc tests was performed for experiments using SPSS software. Significant differences between genotypes were marked by asterisks on graphs with significance at the level of p<0.01 unless otherwise indicated. Error bars on graphs represent standard error of the mean (SEM).

## Acknowledgements

This work was supported by NIH grants P01 DK059820, R01 DK085372, P01 DK0059820, USDA ARS grant 6250-52000-05 and the RP Doherty, Jr Welch Chair in Science (DDM), and NIDDK T32

DK007696 (JLM). We are grateful to Drs Juan Hernandez for help in isolating primary hepatocytes and Rui Xiao for help in generating adenovirus. We thank Dr Richard Whitby for the generous gift of RJW100. We also thank Dr Gerald Thiel for providing the pCMV-Flag-C2/ATF2 vector, Dr Thomas Brunner for providing the EGFP-tagged LRH-1 vector, and Dr Paul Chiao for providing the pCMV-PLK3 vector.

## Additional information

### Funding

| Funder | Grant reference number | Author |
| --- | --- | --- |
| National Institutes of Health | DK059820 | David D Moore |
| National Institutes of Health | DK085372 | David D Moore |
| United States Department of Agriculture | 6250-52000-05 | David D Moore |
| National Institutes of Health | P01 DK0059820 | David D Moore |
| National Institutes of Health | T32DK007696 | Jennifer L Mamrosh |

The funders had no role in study design, data collection and interpretation, or the decision to submit the work for publication.

### Author contributions

JLM, Conception and design, Acquisition of data, Analysis and interpretation of data, Drafting or revising the article; JML, MW, Conception and design, Acquisition of data, Drafting or revising the article; PJS, RJW, S-PW, M-JT, FJDM, Conception and design, Drafting or revising the article, Contributed unpublished essential data or reagents; RNS, DDM, Conception and design, Analysis and interpretation of data, Drafting or revising the article

### Ethics

Animal experimentation: This study was performed in strict accordance with the recommendations in the Guide for the Care and Use of Laboratory Animals of the National Institutes of Health. All of the animals were handled according to approved institutional animal care and use committee (IACUC) at Baylor College of Medicine (protocol AN-4746), and all efforts were taken to minimize pain and distress.

## Additional files

### Supplementary files

• Supplementary file 1. Primer sequences.

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
