## [Decision Letter]

Thank you for sending your work entitled “Nuclear receptor LRH-1/NR5A2 is required and targetable for Endoplasmic Reticulum stress resolution” for consideration at *eLife*. Your article has been favorably evaluated by a Senior editor and 3 reviewers, one of whom is a member of our Board of Reviewing Editors.

The Reviewing editor and the other reviewers discussed their comments before we reached this decision, and the Reviewing editor has assembled the following comments to help you prepare a revised submission.

The reviewers were in agreement that the work was novel, well executed and potentially of interest to the readers of *eLife*. However, several major issues were identified that need to be addressed in order to strengthen the conclusions drawn. In particular, although it is clear that LRH-1 affects ER stress in the liver, concerns were raised as to whether it is appropriate to consider this a new branch of the UPR. Furthermore, additional data on the role of ATF-2 in the LRH-1 pathway seem to be needed as outlined below.

1) The central message of the paper, that this is a novel UPR branch, is premature and not supported with these data and do not reflect an accurate description of the findings. It would be imperative to either demonstrate that the proposed pathway is a new branch of UPR or simply change the title and the central premise of the paper to “identification of a role for LRH-1 in hepatic ER stress”. There is an important difference between these concepts.

2) In essentially all of the experiments examining the role of LRH-1 in the UPR, the authors have examined the levels of XBP1, ATF6, and ATF4, and use these measure as a direct reflection of ER stress through canonical branches. The evidence supporting that there is unresolved ER stress is primarily based on the fact that upon tunicamycin treatment or hepatectomy, the livers of knockout mice exhibit steatosis. It is important to characterize the condition of the ER with additional measures, to come closer to this interpretation. An important aspect of this is the status of the proximal sensors, such as IRE1 and PERK phosphorylation and the evidence for the emergence of maladaptive responses such as apoptotic signals, JNK activation, and others. Otherwise the possibility remains that this ER is under a new equilibrium with the boosted UPR and that LRH-1 is linked to lipid metabolism, rather than representing a bona fide novel UPR branch.

3) Does ER stress result in activation state or subcellular localization of LRH-1? Are there any other indications for direct modification of this molecule by ER stress? If so, is such effect transmitted through IRE1 or PERK?

4) The authors claim that the defective ER stress resolution in LRH-1 liver-specific knockout mice is due to the loss of ATF2 phosphorylation in response to ER stress stimuli. However, the current data do not establish that ATF2 phosphorylation is essential for the ER stress resolution. Could the authors knockdown ATF2 in mice liver and examine whether there is any defects in ER stress resolution after challenge? Alternatively, have the authors attempted to explore whether ATF2- or PLK3-deficient livers exhibit hepatosteosis and/or alterations in ER stress responses?

5) The evidence linking ATF2 activation solely to ER stress response is incomplete, as PLK3 has many other interacting and binding partners. p53, HIF1A, and c-Jun can participate in mediating both cellular stress as well as ER stress. For example, exposure of hepatocytes to toxins such as tunicamycin triggers ER stress, which then feeds into the cellular oxidative stress response. In that case, the ATF2 activation is potentially part of the redox stress response. Are there signalling cascades other than ATF2 upregulated in mice treated with either LRH-1 ligand or exogenous toxins? Comments on this would be helpful. These possibilities need to be addressed by either additional experiments looking at downstream targets of PLK3 or at least in the discussion section of the article.

6) Does LRH-1 ligand affect the phosphorylation of ATF2 in response to ER stress stimuli?

---

## [Author Response]

*1) The central message of the paper, that this is a novel UPR branch, is premature and not supported with these data and do not reflect an accurate description of the findings. It would be imperative to either demonstrate that the proposed pathway is a new branch of UPR or simply change the title and the central premise of the paper to “identification of a role for LRH-1 in hepatic ER stress”. There is an important difference between these concepts*.

We completely agree that the concepts are quite different, and in our initial submission we did not intend to claim that this LRH-1-initiated pathway for ER stress resolution is a fourth branch of the UPR. We have no evidence that it directly contributes to restoration of ER function, for example by promoting protein folding, and we did not observe differential expression of UPR target genes following LRH-1 agonism (Figure 7). In addition, LRH-1 is not universally expressed. Thus, as is now pointed out in the discussion, this pathway cannot be considered equivalent to the three canonical branches. We have also added “*liver* endoplasmic reticulum stress resolution” to the title, as suggested. Nonetheless, the PLK3/ATF2 axis is quite separate from the known UPR components, and we believe that the results strongly support our conclusion that this pathway “is independent of the UPR, yet equivalently required” for stress resolution.

To address the issue of functional interaction with the canonical pathways more directly, we used siRNA to knockdown *Ire1a*, *Perk*, and *Atf6* in primary hepatocytes and determined whether ER stress-responsive induction of LRH-1 target genes was affected. Since the metabolic LRH-1 target genes *Cyp7A1* or *Cyp8B1* are deregulated in cultured cells, we focused on *Shp* and *Plk3.* We observed that both are stress-inducible, but their induction is significantly blunted with loss of *Ire1a* (Figure 3*)*. Since *Ire1a* expression is also critical for induction of *Lrh-1* mRNA (Figure 3), however, it is possible that IRE1a primarily affects the expression of LRH-1 rather than its activity.

*2) In essentially all of the experiments examining the role of LRH-1 in the UPR, the authors have examined the levels of XBP1, ATF6, and ATF4, and use these measure as a direct reflection of ER stress through canonical branches. The evidence supporting that there is unresolved ER stress is primarily based on the fact that upon tunicamycin treatment or hepatectomy, the livers of knockout mice exhibit steatosis. It is important to characterize the condition of the ER with additional measures, to come closer to this interpretation. An important aspect of this is the status of the proximal sensors, such as IRE1 and PERK phosphorylation and the evidence for the emergence of maladaptive responses such as apoptotic signals, JNK activation, and others. Otherwise the possibility remains that this ER is under a new equilibrium with the boosted UPR and that LRH-1 is linked to lipid metabolism, rather than representing a bona fide novel UPR branch*.

We have obtained new data to support our conclusion that ER stress is sustained and toxic in mice lacking *Lrh-1*. While we previously showed that *Lrh-1*^*LKO*^ mice exhibit increased cell death following tunicamycin by TUNEL staining, we have now also investigated caspase activation following stress in our mice. As expected, we observed greater stress-induced caspase activation in *Lrh-1*^*LKO*^ mice (Figure 1).

We also used thioflavin T, which fluoresces when bound to protein aggregates (Beriault and Werstuck, 2013) as an additional method to visualize and quantify ER stress. In tunicamycin treated primary hepatocytes from control and *Lrh-1*^*LKO*^ mice, we observed strikingly increased fluorescence at later time points following tunicamycin treatment in *Lrh-1*^*LKO*^ mice (Figure 1). This indicates that loss of *Lrh-1* not only affects UPR signaling components, but also contributes to greater protein misfolding.

We do not believe that the ER in *Lrh-1*^*LKO*^ mice is at a new equilibrium following stress for the following reasons: 1) Thioflavin T staining indicates a much greater quantity of protein aggregates in *Lrh-1*^*LKO*^ mice following ER stress (Figure 1). 2) We do not observe any UPR activation in *Lrh-1*^*LKO*^ mice prior to treatment with ER stressors (Figure 1; Figure 1—figure supplement 1). 3. We observe greater apoptosis in response to a wide range of stressors in *Lrh-1*^*LKO*^ mice (Figure 1; Figure 2).

While we cannot exclude that the steatosis observed *Lrh-1*^*LKO*^ mice following ER stress could also be the cause for ER stress, we think it is important to note that this fat accumulation is only observed at later time points (48 hr and beyond) post tunicamycin treatment (Figure 1). The UPR responds rapidly (within a few hours) to misfolded proteins and we have shown that our pathway is maximally activated within several hours following treatment with tunicamycin. Therefore, the bulk of the ER stress response (and, one could argue, resolution) in normal mice occurs much before accumulation of significant amounts of fat in mice deficient in ER stress resolution. While it is possible that LRH-1 is critical for aspects of lipid metabolism, it is also important to note that lipids are not significantly increased in *Lrh-1*^*LKO*^ mice prior to ER stress (Figure 1). Therefore, we maintain that LRH-1 is critical for ER stress resolution.

*3) Does ER stress result in activation state or subcellular localization of LRH-1? Are there any other indications for direct modification of this molecule by ER stress? If so, is such effect transmitted through IRE1*
*or PERK?*

In new results we show that ER stress does not affect subcellular localization; LRH-1 is exclusively nuclear with or without ER stress (Figure 3). Thus, LRH-1 is unlikely to be directly phosphorylated or modified by ER-bound UPR factors.

We found that induction of LRH-1 target genes is increased following ER stress, which may be due to increased expression of *Lrh-1* mRNA. We found (as detailed in point 1 above) that IRE1a and ATF6 may control this induction of *Lrh-1*. However, a more intriguing question is whether the transcriptional activity of LRH-1 is increased following ER stress, potentially via induction of an agonist ligand or post-translational modification. Unfortunately, both are very difficult to address as nothing is known regarding endogenous LRH-1 agonists, and little is known about regulation of LRH-1 activity in any context.

*4) The authors claim that the defective ER stress resolution in LRH-1 liver-specific knockout mice is due to the loss of ATF2 phosphorylation in response to ER stress stimuli. However, the current data do not establish that ATF2 phosphorylation is essential for the ER stress resolution. Could the authors knockdown ATF2 in mice liver and examine whether there is any defects in ER stress resolution after challenge? Alternatively, have the authors attempted to explore whether ATF2- or PLK3-deficient livers exhibit hepatosteosis and/or alterations in ER*
*stress responses?*

We agree that data on the requirement for ATF2 in ER stress resolution would strengthen our story. To address this, we generated adenoviruses expressing a constitutively active ATF2 or a dominant negative ATF2 that cannot be phosphorylated at critical residues. We observed that expression of constitutively active ATF2 in *Lrh-1*^*LKO*^ cells increases their ability to resolve ER stress; conversely, expression of the dominant negative ATF2 in control cells diminishes their ability to resolve ER stress (Figure 6). While it is possible that PLK3 does not exert its protective effect in ER stress resolution solely through ATF2, it is clear that ATF2 (and potentially redundant ATF-family transcription factors) do play critical roles in ER stress resolution.

In response to the later questions, we did find that *Plk3*-deficient livers exhibit sustained UPR signaling following tunicamycin treatment (Figure 5), suggesting that they cannot resolve ER stress as efficiently. We did not observe increased hepatosteatosis due to the fact that the background of this mouse strain is more sensitive to tunicamycin than controls for the *Lrh-1*-deficient mice, and therefore even *Plk3* wild-type mice also exhibit increased hepatosteatosis following TM relative to C57/Bl6 mice. Under basal conditions, however, the PLK3 knockouts do not exhibit either activation of the UPR or significantly elevated triglycerides.

*5) The evidence linking ATF2 activation solely to ER stress response is incomplete, as PLK3 has many other interacting and binding partners. p53, HIF1A, and C-Jun can participate in mediating both cellular stress as well as ER stress. For example, exposure of hepatocytes to toxins such as tunicamycin triggers ER stress, which then feeds into the cellular oxidative stress response. In that case, the ATF2 activation is potentially part of the redox stress response. Are there signalling cascades other than ATF2 upregulated in mice treated with either LRH-1 ligand or exogenous toxins? Comments on this would be helpful. These possibilities need to be addressed by either additional experiments looking at downstream targets of PLK3 or at least in the discussion section of the article*.

It is true that PLK3 has additional interacting partners, including other transcription factors. We have supplemented our manuscript with additional analysis of our array data to determine whether genes differentially induced by stress in *Lrh-1*^*LKO*^ mice could be targets of additional transcription factors known to be phosphorylated by PLK3. This information is summarized in Table 2 and discussed in detail in the Results and Discussion.

Following this additional analysis, the most significant overlap of our gene set (Table 1) with known transcription factor promoter binding remained that for ATF2. However, we also observed some enrichment for p53 and c-Jun target genes (as well as for LRH-1, but this is not thought to be regulated by PLK3). Interestingly, the genes in our gene set that are direct targets of p53 or c-Jun are also direct targets of ATF2. In other words, we did not identify any genes bound by p53 or c-Jun that are not bound by ATF2 in the same region (data not shown in paper for size considerations). Therefore, these studies fail to show evidence for a critical role of p53 or c-Jun in induction of these genes. The fact that expression of a constitutively active ATF2 in *Lrh-1*^*LKO*^ cells facilitates ER stress resolution (Figure 6) argues that ATF2 is a major functional output of PLK3 in ER stress resolution. It is possible that p53 or c-Jun may be forming complexes with ATF2 in the context of ER stress to maximize gene induction. Yet ATF2 is known to bind additional genes in our list that do not appear to be p53 or c-Jun targets, suggesting that ATF2 may work alone (or at least without these transcription factors). Since HIF1a is phosphorylated by PLK3, but this phosphorylation is destabilizing (Xu D, Yao Y, Lu L, Costa M, Dai W. 2010. Plk3 functions as an essential component of the hypoxia regulatory pathway by direct phosphorylation of HIF-1alpha. The Journal of Biological Chemistry 285:38944–38950. doi: 10.1074/jbc.M110.160325.), we did not investigate HIF1a further.

We also investigated whether the oxidative stress response was induced in our target gene list. There are no known oxidative stress transcription factors known to be phosphorylated by PLK3 (unless you include p53), so we compared our gene list to genome wide binding of NRF2. We did not observe significant overlap. At later time points (such as 48 or 72 h) post treatment with tunicamycin, we might expect to observe oxidative stress in the *Lrh-1*^*LKO*^ mice resulting from their unresolved ER stress. However, we can conclude that LRH-1 does not appear to control an oxidative stress response to tunicamycin at earlier time points that constitute the primary ER stress response.

Whether additional, broader pathways are controlled by LRH-1 and/or PLK3 in response to stress remains an open question. However, we had previously performed a variety of non-biased bioinformatics analyses (motif discovery, gene ontology, etc) and have not found anything else significant beyond CRE binding factors/ATF2 (significant data in paper, some data not shown) that would suggest additional pathways dependent on LRH-1 are required for ER stress resolution.

*6) Does LRH-1 ligand affect the phosphorylation of ATF2 in response to*
*ER stress stimuli?*

We have unfortunately not been able to address this directly, due to difficulties with later batches of the phospho-ATF2 antibody (Cell Signaling 9225S). All experiments shown were performed using batch 3 of this antibody, which detects a single clean band. However, now we can only obtain batch 4, which detects multiple bands with no clear pATF2 band. We have not been able to detect pATF2 reliably using other antibodies. We of course hope that this issue is resolved soon for the benefit of our lab and others.

That being said, we would expect the ligand to induce phosphorylation of ATF2 since it induces *Plk3* and also ATF2 target genes, which should be dependent on such phosphorylation (Figure 7).